# The calcium channel subunit α₂δ-3 organizes synapses via an activity-dependent and autocrine BMP signaling pathway

Kendall M. Hoover [1], Scott J. Gratz[2], Nova Qi[1], Kelsey A. Herrmann[1], Yizhou Liu[1], Jahci J. Perry-Richardson[1], Pamela J. Vanderzalm [3], Kate M. O'Connor-Giles[2] & Heather T. Broihier [1]*

Synapses are highly specialized for neurotransmitter signaling, yet activity-dependent growth factor release also plays critical roles at synapses. While efficient neurotransmitter signaling relies on precise apposition of release sites and neurotransmitter receptors, molecular mechanisms enabling high-fidelity growth factor signaling within the synaptic micro-environment remain obscure. Here we show that the auxiliary calcium channel subunit α₂δ-3 promotes the function of an activity-dependent autocrine Bone Morphogenetic Protein (BMP) signaling pathway at the Drosophila neuromuscular junction (NMJ). α₂δ proteins have conserved synaptogenic activity, although how they execute this function has remained elusive. We find that α₂δ-3 provides an extracellular scaffold for an autocrine BMP signal, suggesting a mechanistic framework for understanding α₂δ's conserved role in synapse organization. We further establish a transcriptional requirement for activity-dependent, autocrine BMP signaling in determining synapse density, structure, and function. We propose that activity-dependent, autocrine signals provide neurons with continuous feedback on their activity state for modulating both synapse structure and function.

[1] Department of Neurosciences, Case Western Reserve University School of Medicine, Cleveland, OH 44106, USA. [2] Department of Neuroscience, Brown University, Providence, RI 02912, USA. [3] Department of Biology, John Carroll University, University Heights, OH 44118, USA. *email: heather.broihier@case.edu

Synapses are asymmetric cellular junctions underlying directional information transfer in the nervous system. Activity-dependent pathways are often invoked as regulators of synapse development, maintenance, plasticity, and elimination. While roles for activity-dependent cues at synapses are indisputable, the molecular links between activity and synapse differentiation remain poorly defined. To establish these connections, it is imperative to define when and where activity-dependent cues are released, and then how these extracellular signals impinge on synapse morphology and function.

Classical morphogens and neurotrophins have emerged as central regulators of synaptic development. A diverse collection of secreted cues promote differentiation of the pre- or postsynaptic compartment[1–6]. Such cues are proposed to coordinate pre- and postsynaptic development by signaling across the synapse. Yet autocrine signals can also serve important functions at the synapse by providing ongoing feedback to pre- or postsynaptic cells regarding their activity state[7]. As increasing numbers of synaptic signals are identified, the question arises of how neurons discriminate among these information channels. It is reasonable to hypothesize that proteins in the synaptic cleft serve crucial roles controlling the spatiotemporal availability of secreted proteins to facilitate signal segregation.

$\alpha_2\delta$ proteins are well-positioned to impact the activity of synaptic signals. They are best studied as accessory calcium channel subunits and are composed of two disulfide-linked peptides: an entirely extracellular $\alpha_2$ peptide and a $\delta$ peptide predicted to be membrane-associated via a GPI link[8,9]. They are important for voltage-gated calcium ($Ca_V$) channel biophysical properties and trafficking[8,10,11]. Interestingly, presynaptic $\alpha_2\delta$-2 has also been shown to promote $GABA_A$ receptor levels[12]. Underscoring $\alpha_2\delta$ functions at mature synapses, $\alpha_2\delta$ proteins are cellular targets of Gabapentin[13], a commonly prescribed treatment for epilepsy and neuropathic pain. All $\alpha_2\delta$ proteins contain von Willebrand factor A (VWA) domains, an interaction motif characteristic of ECM proteins such as matrilins and collagens[14]. $\alpha_2\delta$ proteins interact with synaptogenic proteins including Thrombospondin-1 (refs. [15,16]), suggesting functions beyond $Ca_V$ channel localization and function. In line with this hypothesis, Drosophila $\alpha_2\delta$-3, also called Straightjacket (Stj), promotes bouton morphogenesis at the embryonic neuromuscular junction (NMJ)—a function independent of the $Ca_V$ channel Cacophany (Cac)[17]. Functions for Drosophila $\alpha_2\delta$-3 in subsequent aspects of NMJ differentiation are incompletely understood, but are reported to include synaptic assembly, synaptic function, and presynaptic homeostatic plasticity[18–20].

Here we report a functional requirement for Drosophila $\alpha_2\delta$-3 in activity-dependent autocrine BMP signaling. At the NMJ, postsynaptic Gbb (Glass bottom boat), a BMP family member, is an early and permissive retrograde cue that initiates scaling growth of the NMJ[21,22]. We previously demonstrated that presynaptic Gbb is also trafficked to presynaptic terminals, where it is released following activity[23,24]. What is the function of this activity-dependent cue? We show that it is an autocrine signal maintaining synapse structure and function. In the absence of presynaptic Gbb signaling, active zone architecture, synaptic vesicle distribution, and baseline neurotransmitter release are all disrupted. Unexpectedly, embryonic bouton morphogenesis is also disrupted, a phenotype identical to that seen in $\alpha_2\delta$-3 mutants[17], prompting us to test for a link between BMP signaling and $\alpha_2\delta$-3. We provide evidence that the extracellular $\alpha_2$ peptide of $\alpha_2\delta$-3 promotes membrane retention of Gbb following its activity-dependent release. We therefore propose that $\alpha_2\delta$-3 is a key component of the synaptic cleft microenvironment serving to limit the diffusion of extracellular Gbb.

## Results

### A presynaptic and autocrine BMP pathway at the NMJ.
Classic studies demonstrate that BMP signaling orchestrates NMJ development and physiology in Drosophila. Loss of BMP signaling causes a reduction in NMJ size as judged by bouton number—as well as ultrastructural defects, reduced evoked glutamate release, and impaired homeostatic plasticity[21,25–27]. These widespread defects raise the question of whether the phenotypes have a common root or, if instead they reflect separable, cell type-specific roles for BMP signaling. Early work suggested at least partially separable pre- and postsynaptic BMP pathways; while expression of Gbb in the postsynaptic muscle rescues bouton number in gbb nulls, it does not rescue evoked neurotransmitter release. Expression of Gbb in the presynaptic neuron is required to restore proper glutamate release[21,24,26]. These findings suggest that Gbb is released by presynaptic motor neurons. Lending key support to this idea, Gbb is trafficked to presynaptic terminals, where it is subject to activity-dependent release[24].

We hypothesized that this presynaptic pool of Gbb regulates synapse formation or maintenance. To explore this idea, we first established the effect of complete loss of Gbb on synapses. As expected, bouton number is significantly decreased in gbb nulls (Supplementary Fig. 1A–B, H). Each bouton contains many synapses, or individual presynaptic glutamate release sites precisely aligned to postsynaptic glutamate receptor clusters. We utilized the ELKS-related protein Bruchpilot (Brp) as a presynaptic marker, GluRIII as a postsynaptic marker—and defined a synapse as a pair of Brp/GluRIII puncta[28–31]. We scored Brp-positive synapse density (synapse number per $\mu m^2$) to exclude differences in synapse number arising as a secondary consequence of altered overall NMJ size. Loss of Gbb results in a 30% decrease in Brp-positive synapse density (Fig. 1a, b, i), indicating that Gbb regulates synapse development.

We defined the relevant cellular source of Gbb by selectively removing either pre- or postsynaptic ligand. To test necessity of the postsynaptic pool we used muscle-specific gbb RNAi ($gbb^1/+$; BG57>gbb RNAi). In both this RNAi experiment and those described below, knockdown was performed in gbb heterozygotes, which does not impact synapse density in an otherwise wild-type background (Fig. 1i). To confirm the RNAi-based approach and to test sufficiency of the presynaptic pool, we overexpressed gbb in motor neurons in gbb nulls ($gbb^{1/2}$; D42>Gbb). As expected, muscle-derived ligand regulates bouton number[21,27] (Supplementary Fig. 1A, C, H). In contrast, we do not detect a change in Brp/GluRIII density (Fig. 1a, c, i), indicating that muscle-derived Gbb does not otherwise regulate synapse development. We gained genetic access to motor neuron-derived Gbb using analogous strategies. We utilized motor neuron-specific RNAi ($gbb^1/+$; D42>gbb RNAi) to probe a requirement for presynaptic ligand, while to test sufficiency of the postsynaptic ligand, we expressed gbb in muscle in gbb nulls ($gbb^{1/2}$; BG57>Gbb). Bouton number is unchanged in these backgrounds (Supplementary Fig. 1A, D, H). However, the density of Brp/GluRIII puncta in both backgrounds is decreased comparably to gbb nulls (Fig. 1a, d, i). Thus, the synapse density phenotype observed in gbb nulls is attributable to neuron-derived ligand.

Wishful thinking (Wit) is the Type II BMP receptor mediating Gbb function in NMJ growth[25,27]. We were curious if Wit is required for Gbb's synapse density function. As expected, loss of Wit results in fewer boutons (Supplementary Fig. 1A, E, H). In addition, we observe a 40% decrease in Brp-positive synapse density (Fig. 1a, e, i). To test for a requirement of presynaptic Wit, we assayed whether Wit expression in motor neurons in wit nulls ($wit^{A12/B11}$; D42>Wit) rescues the synapse density phenotype. Indeed, synapse density is fully rescued in this background (Fig. 1a, e, f, i). The genetic requirement for ligand and receptor

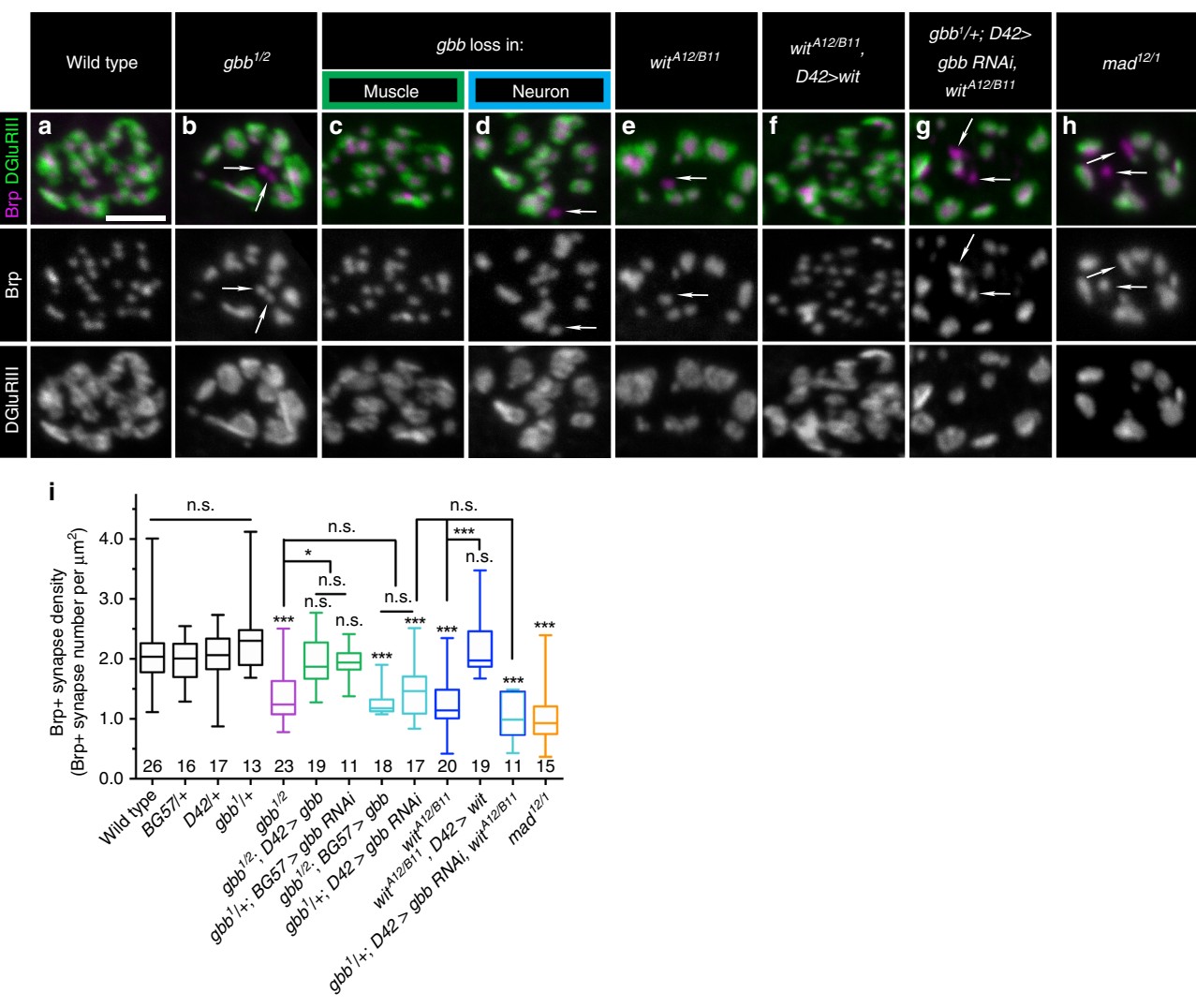

**Fig. 1 A presynaptic and autocrine BMP pathway at the NMJ. a–h** Representative *z*-projections of boutons of indicated genotypes labeled with Brp (magenta) and DGluRIII (green). *gbb* loss in muscle is *gbb$^{1/2}$; D42>gbb*, and *gbb* loss in neuron is *gbb$^{1/2}$; BG57>gbb*. Arrows denote unapposed Brp+puncta. Individual channels are shown in grayscale below or next to merged images for clarity throughout all figures. Scale bar: 2 μm. **i** Quantification of Brp+ synapse density (number of Brp+ synapses per terminal bouton area in square microns). *n* is the number of boutons scored. Error bars are min and max data points, and the center line indicates the median. n.s. not significantly different. *$p < 0.05$; ***$p < 0.001$. Statistical test is a nonparametric Kruskal–Wallis one-way ANOVA on ranks followed by Dunn's multiple comparison test. For all figures, source data are provided as a Source Data file.

in the presynaptic neuron suggested an autocrine BMP signaling loop. The prediction for a such a pathway is that the double mutant phenotype would be no stronger than that of either single mutant, so we sought to analyze synapse density in *gbb; wit* double homozygotes. However, the double mutants were late-embryonic lethal. Instead, we analyzed synapse density in animals with presynaptic *gbb* knockdown in *wit* nulls (*gbb$^{1}$/+; D42>gbb RNAi, wit$^{A12/B11}$*). We do not detect a further decrease in synapse density in these animals (Fig. 1d, e, g, i), supporting an autocrine pathway. Lastly, we tested if this pathway requires the transcription factor Mad. Loss of Mad results in a 45% reduction in synapse density (Fig. 1a, h–i), supporting a canonical BMP pathway. Together, these findings uncover a canonical, presynaptic, and autocrine BMP pathway that promotes Brp-positive synapse density at the NMJ.

**Presynaptic BMP signaling regulates active zone architecture.**
In the preceding analysis, we had the impression that loss of autocrine BMP signaling results in larger Brp puncta (Fig. 1).

When labeled with Brp antibody and imaged in planar orientation at higher resolution, Brp puncta resolve into roughly 200 nm diameter rings (Fig. 2a)[29]. Loss of active zone cytomatrix components alters Brp ring geometry[32–36] making it a sensitive probe for assembly of the highly-ordered cytomatrix. Thus, we analyzed Brp ring size and distribution utilizing extended resolution microscopy.

In controls, we find that the average diameter of isolated Brp rings is roughly 200 nm, consistent with published reports (Fig. 2a, i–k; Supplementary Fig. 2A–C′)[28,32–35]. In contrast, mean Brp ring diameter increases by 23% in *gbb* nulls (Fig. 2a, b, k). We investigated the relevant cellular source of Gbb using cell-type-specific knockdown and rescue genotypes as described above. Loss of neuronal Gbb results in increased Brp ring diameter analogous to that seen in *gbb* nulls, while loss of muscle-derived Gbb has no effect (Fig. 2a–d, k). We next analyzed if Wit and Mad regulate cytomatrix architecture and find that loss of either results in increased Brp ring diameter (Fig. 2a, e, h, k). The *wit* phenotype is rescued by neuronal expression of Wit (Fig. 2a, e–f, k). Consistent with a linear pathway, knockdown of

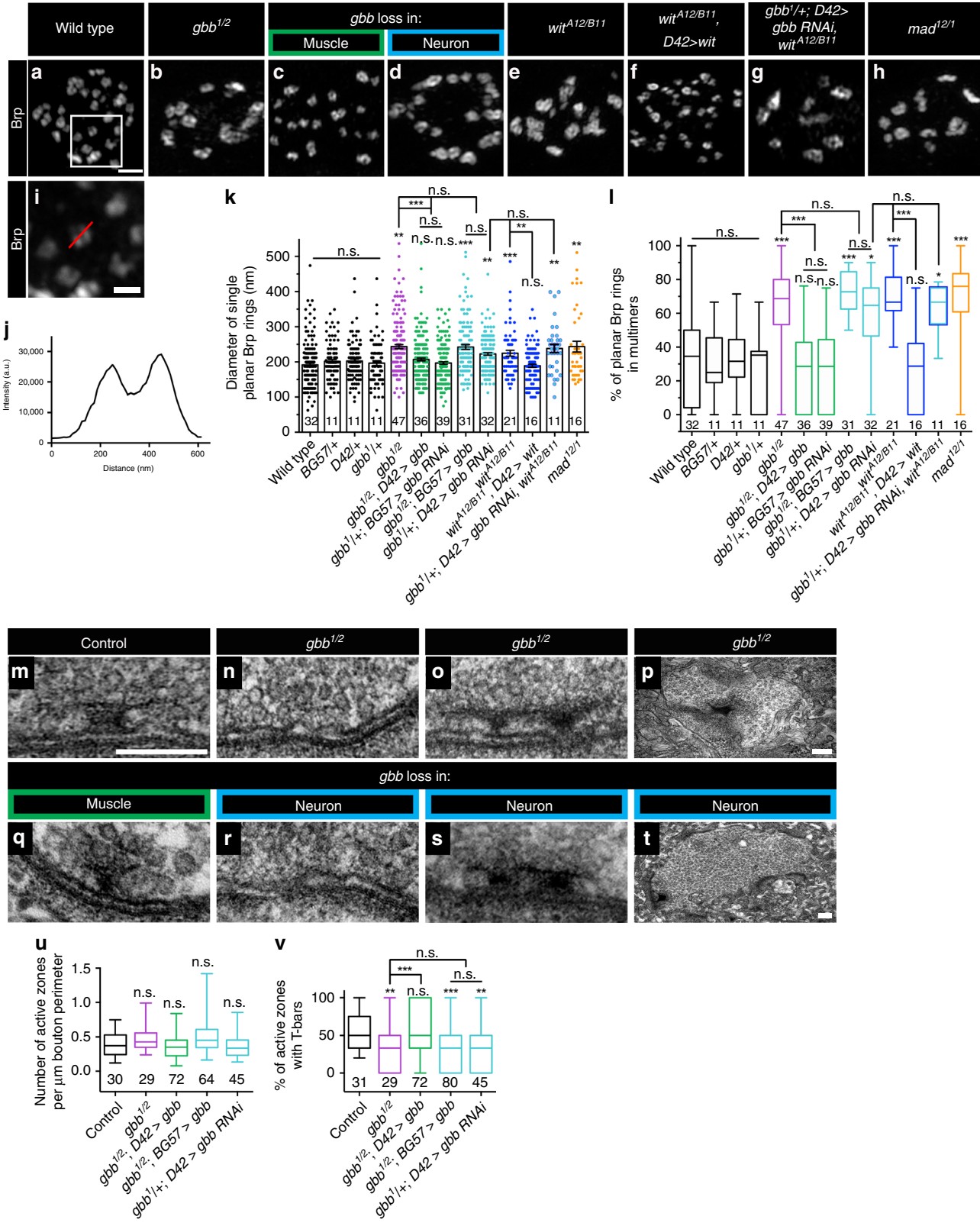

presynaptic Gbb in a *wit* null background does not exacerbate the phenotype of either genetic manipulation on its own (Fig. 2d, e, g, k). The increased Brp ring diameter in autocrine BMP mutants indicates that the pathway ensures precise construction and/or maintenance of the active zone cytomatrix. Notably, increased Brp ring size is mirrored by increased glutamate receptor field

size (Supplementary Fig. 1i), suggesting that presynaptic Gbb signaling regulates postsynaptic differentiation.

We also noticed a marked defect in the distribution of Brp rings in autocrine BMP pathway mutants. In controls, the majority of Brp rings are observed as isolated rings, while in autocrine BMP mutants, many rings are present in

**Fig. 2 Presynaptic BMP signaling regulates active zone architecture. a–h** Representative deconvolved z-projections of boutons of indicated genotypes labeled with anti-Brp. gbb loss in muscle is gbb$^{1/2}$; D42>gbb, and gbb loss in neuron is gbb$^{1/2}$;BG57>gbb. Scale bar: 1 μm. **i** Zoomed in view of the box in **a**, demonstrating how Brp ring diameter was measured. The red line is typical of a manually drawn line through a single planar Brp ring. Scale bar: 200 nm. **j** Fluorescence intensity along the line in **i** was plotted, and Brp ring diameter was calculated as the distance between the two intensity maxima. **k** Quantification of single planar Brp ring diameters. **l** Quantification of the percentage of planar Brp rings within multimers, or groups of interconnected Brp rings. For all Brp ring analyses, n is the number of boutons scored. **m–t** Representative transmission electron micrographs of active zones (**m–o**, **q–s**) or boutons (**p, t**) of the indicated genotypes. gbb loss in muscle is gbb$^{1/2}$;D42>gbb, and gbb loss in neuron is gbb$^{1/2}$;BG57>gbb. Scale bars: 200 nm. **u** Quantification of the number of active zones per micron bouton perimeter. n is the number of boutons scored. **v** Quantification of the percentage of active zones with T-bars. n is the number of active zones scored. For the bar graph, error bars are mean ± SEM. For all box-and-whisker plots, error bars are min and max data points, and the center line indicates the median. Individual data points are displayed as dots. n.s. not significantly different. *$p < 0.05$; **$p < 0.01$; ***$p < 0.001$. All tests are nonparametric Kruskal–Wallis one-way ANOVAs on ranks followed by Dunn's multiple comparison test.

interconnected, disorganized clusters. We quantified this phenotype and found that roughly 31% of Brp rings are interconnected in controls, relative to 66% in gbb nulls (Fig. 2a, b, l). This phenotype is apparent when presynaptic, but not postsynaptic, ligand is lost (Fig. 2a, c, d, l), and likewise requires Mad and neuronal Wit (Fig. 2a, e, f, h, k). The penetrance of the interconnected Brp ring phenotype is not enhanced by simultaneous loss of Wit and presynaptic Gbb (Fig. 2d, e, g, l), again supporting a linear pathway. These analyses suggest that autocrine BMP signaling regulates Brp allocation among active zones.

Together, these light-level analyses suggest that the decrease in Brp-positive synapse density in autocrine BMP mutants may be explained by altered Brp distribution among active zones. In other words, some active zones may have excess Brp while others have none. Brp is an essential component of electron-dense T-bars visible at the EM level[29]. Thus, BMP pathway mutants might exhibit altered T-bar distribution at the ultrastructural level. We first asked if Gbb regulates the number of synapses, here defined as electron-dense tightly apposed pre- and postsynaptic membranes, and did not detect any alterations (Fig. 2u). Presynaptic Gbb does promote adhesion of pre- and postsynaptic membranes, as membrane ruffling increases in mutant backgrounds (from <5% in controls to roughly 20% in mutants).

As predicted, we find a reduction in the percentage of active zones containing T-bars in presynaptic BMP mutants. In our electron micrographs, 59% of control active zones have visible T-bars, in line with published work (Fig. 2m, v)[37]. In contrast, T-bars are present at only 32% of gbb null active zones (Fig. 2n, v). This phenotype tracks specifically with loss of the presynaptic ligand pool (Fig. 2r, v). We observed two additional phenotypes in presynaptic Gbb mutants. First, we find an increase in active zones with multiple T-bars (Fig. 2o, s). And second, we observe membrane-detached aggregates of electron-dense material (Fig. 2p, t), which were never observed in controls, but have been reported with constitutive loss of the BMP pathway[21,38]. We propose that this phenotype contributes to the unopposed Brp puncta observed at the light level (arrows in Fig. 1). Together, these analyses indicate that autocrine BMP signaling regulates the architecture and composition of the presynaptic compartment.

**Presynaptic BMP signaling regulates SSV distribution.** Might BMP signaling regulate additional features of the presynaptic compartment? We investigated small synaptic vesicle (SSV) localization as tight regulation of vesicle number and distribution underpins presynaptic function by analyzing SSV distribution using the vesicular glutamate transporter (VGLUT)[39]. SSVs cluster at active zones and are enriched at presynaptic membranes, leaving bouton interiors relatively less populated with SSVs (Fig. 3a). In gbb mutants, SSV distribution appeared more uniform (Fig. 3b). To quantify this phenotype, we assessed relative SSV distribution by drawing a line through the bouton center

and analyzing the fluorescence intensity profile along the line. We defined proper SSV distribution as corresponding to a 50% decrease in VGLUT fluorescence intensity in the bouton interior (Fig. 3i–j) and analyzed only boutons of at least 2 μM diameter to avoid small boutons lacking well-defined SSV distributions. We first calculated integrated VGLUT fluorescence intensity as a proxy for overall SSV number and find no alterations in any mutant background (Fig. 3l).

Turning to SSV distribution, while 82% of control boutons have peripherally localized SSVs, 29% of boutons in gbb nulls exhibit proper SSV distribution (Fig. 3a, b, k). Presynaptic, and not postsynaptic, Gbb is required for normal SSV distribution within boutons (Fig. 3a, c, d, k). Interestingly, a similar requirement for neuronal Gbb in SSV distribution was suggested by our EM analysis (Supplementary Fig. 2D–G). We asked whether neuronal Wit and Mad are likewise involved. Supporting a function for Wit, only 21% of boutons in wit nulls have peripheral SSV distribution. This phenotype is rescued by neuronal expression of Wit (Fig. 3a, e–f, k), and is no more severe with simultaneous loss of Wit and presynaptic Gbb (Fig. 3d, e, g, k). Loss of Mad results in a slightly more intermediate phenotype (Fig. 3a, h, k). These findings argue that presynaptic, autocrine Gbb signaling regulates SSV distribution.

Notably, presynaptic phenotypes akin to those observed in presynaptic Gbb pathway mutants have been found in neurexin-1 (nrx-1) and syd-1 mutants. Loss of either nrx-1 or syd-1 is reported to lead to fewer and larger Brp scaffolds[32,33,40], and loss of nrx-1 is reported delocalize SSVs[41]. To facilitate a direct comparison of the functions of nrx-1 and syd-1 with the presynaptic BMP pathway described here, we assayed synapse density, Brp ring size, and SSV distribution in nrx-1$^{241/273}$ and syd-1$^{ex1.2/ex3.4}$ mutants. In our hands, loss of Nrx-1 results in somewhat similar phenotypes to BMP pathway mutants, while loss of Syd-1 has less pronounced effects (Supplementary Fig. 3A–M).

**Presynaptic Gbb is necessary for baseline synaptic function.** Genetic rescue experiments establish that neuronal Gbb is sufficient to restore normal baseline glutamate release to gbb nulls[21,24,26]. While these studies demonstrate that neuronal Gbb can support normal transmission, they do not demonstrate that neuronal Gbb normally does serve this function. The abnormalities in active zone structure and vesicle distribution that we observe with loss of neuronal Gbb support a role for this ligand pool in transmission.

To address this question directly, we measured spontaneous and evoked synaptic potentials via intracellular recordings in gbb RNAi backgrounds. We find that mutants with Gbb removed selectively from motor neurons exhibit a 53% reduction in evoked EJP amplitude, while mutants with Gbb removed from the muscle have a 26% reduction (Fig. 4a–e). The 26% reduction in EJP amplitude with muscle-specific Gbb knockdown was expected as

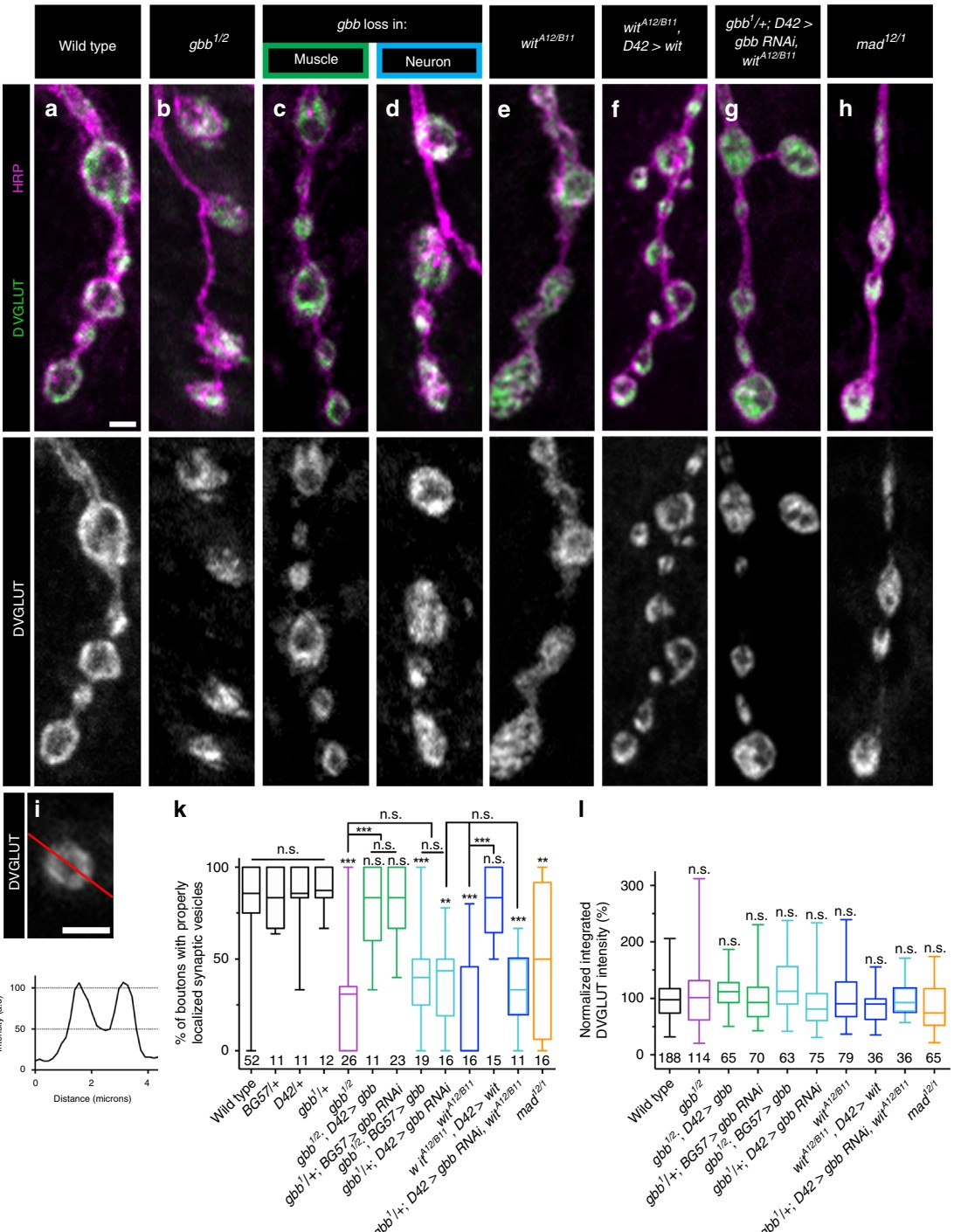

**Fig. 3 Presynaptic BMP signaling regulates SSV distribution. a–h** Representative z-projections of boutons of the indicated genotypes labeled with DVGLUT (green) and HRP (magenta). *gbb* loss in muscle is *gbb*$^{1/2}$; *D42>gbb*, and *gbb* loss in neuron is *gbb*$^{1/2}$; *BG57>gbb*. Scale bar: 2 μm. **i** The red line represents a typical line drawn through a bouton larger than 2 μm in diameter. Scale bar: 2 μm. **j** The fluorescence intensity graph along the line (**i**) was created as shown. If the two intensity maxima were each greater than twice the minimum, the bouton had properly distributed synaptic vesicles.
**k** Quantification of the percentage of boutons properly distributed synaptic vesicles. For all DVGLUT localization analyses, *n* is the number of NMJs scored.
**l** Quantification of integrated DVGLUT intensity normalized to controls. For all integrated DVGLUT analyses, *n* is the number of boutons scored. Error bars are min and max data points, and the center line indicates the median. n.s. not significantly different. **\*\*p < 0.01; \*\*\*p < 0.001**. All tests are nonparametric Kruskal–Wallis one-way ANOVAs on ranks followed by Dunn's multiple comparison test.

these NMJs display a 22% reduction in overall NMJ size without increased synapse density (Fig. 1, Supplementary Fig. 1). In contrast, the 53% reduction in EJP amplitude with loss of neuron-specific Gbb knockdown cannot be explained by a change in gross NMJ morphology since bouton number is unchanged in this background (Supplementary Fig. 1A, D, H). Instead, these findings support a specific requirement for presynaptic Gbb in synaptic function. Indeed, quantal content is significantly reduced upon presynaptic loss of Gbb (Fig. 4g), indicating aberrant evoked glutamate release consistent with the observed structural

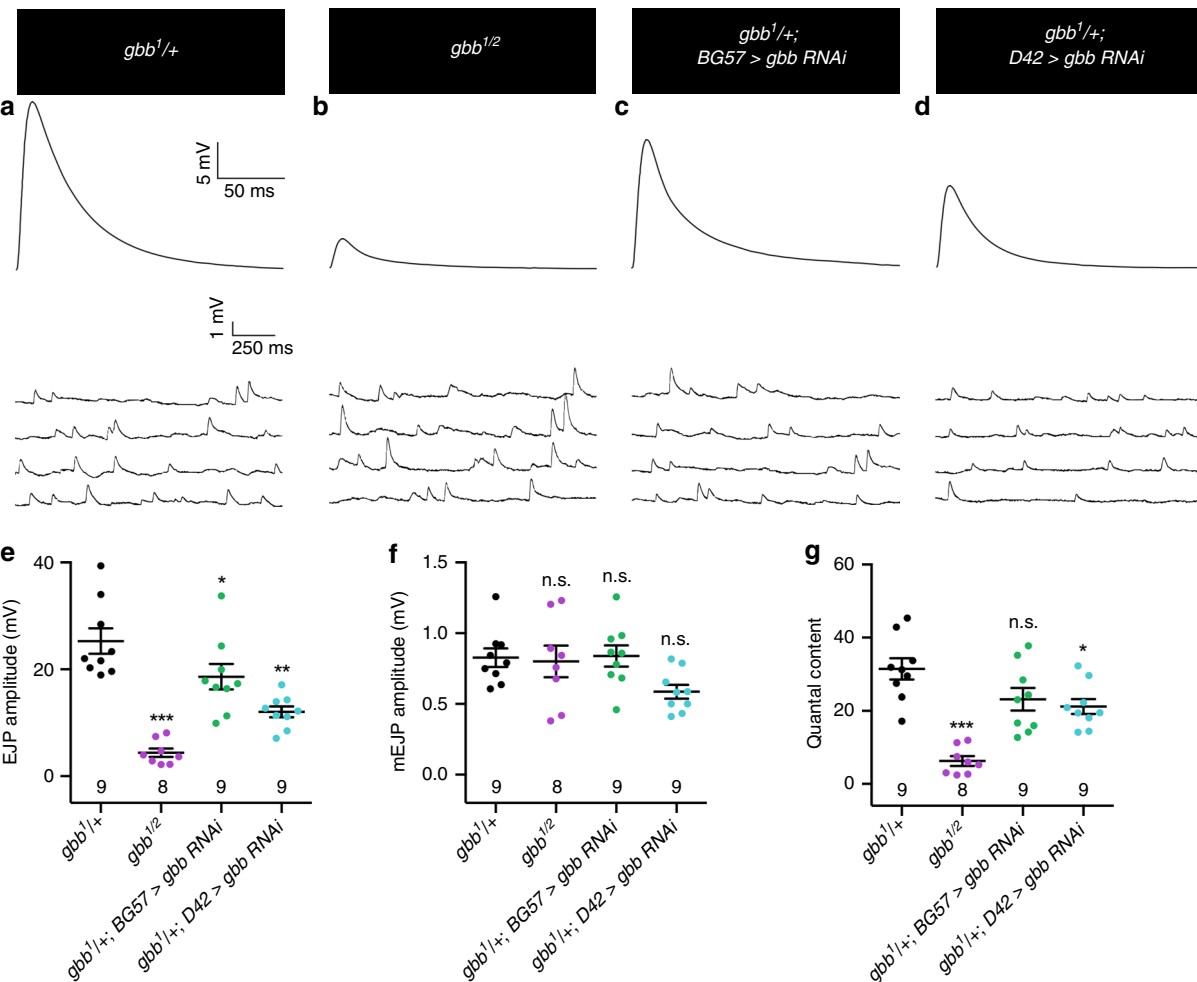

**Fig. 4 Presynaptic Gbb is necessary for baseline synaptic function. a–d** Representative EJP (upper) and mEJP (lower) traces for the indicated genotypes. **e** Quantification of EJP amplitude. **f** Quantification of mEJP amplitude. **g** Quantal content for each genotype was calculated by dividing average EJP amplitude by average mEJP amplitude. For all electrophysiological experiments, $n$ is the number of cells recorded. Error bars are mean ± SEM. Individual data points are displayed as dots. n.s. not significantly different. *$p < 0.05$; **$p < 0.01$; ***$p < 0.001$. All tests are one-way ANOVAs followed by Dunnett's post hoc test.

deficits. These findings extend previous genetic rescue data[21,24,26] to establish a functional requirement for presynaptic Gbb in evoked neurotransmitter release.

**An ongoing requirement for presynaptic BMP signaling.** Gbb and Mad are required exclusively during the first instar larval stage for NMJ growth[22], arguing for an early critical period for the retrograde pro-growth cue. To define the temporal requirement of presynaptic Gbb, we utilized the GeneSwitch–Gal4 system (GS–Gal4) for inducible expression of Gbb in all neurons via elav-GS-Gal4[42]. We first tested if proper synapse density is rescued by inducing Gbb in neurons during the second instar larval stage (L2) and find that it is restored to wild-type levels (Fig. 5a, b, d). Moreover, later induction of Gbb at third instar (L3) fully rescues the *gbb* null phenotype (Fig. 5a, c, d). Supporting the temporal distinction between an early pro-growth signal and a later synapse-organizing signal, bouton number is not rescued with L2 or L3 Gbb induction (Fig. 5e). We also measured planar Brp ring diameter in *gbb; Elav-GS>Gbb* animals and find that inducing Gbb at either L2 or L3 restores wild-type Brp ring diameter (Fig. 5f–h, i). Finally, the interconnected Brp ring phenotype of *gbb* nulls is also completely rescued by L2 or L3 induction of Gbb (Fig. 5f–h, j). These findings are consistent with prior work indicating a later larval requirement for Mad in neurotransmission[22] and imply that

inducing autocrine BMP signaling after synapses have formed corrects aberrant synapse structure.

The separable temporal requirements raise the possibility that the autocrine pathway is the predominant BMP pathway active at the L3 NMJ. In this event, autocrine signaling might more efficiently drive BMP signal transduction at this stage. We tested this hypothesis by comparing the effectiveness of pre- and postsynaptic ligand to drive nuclear pMad localization. As anticipated, pMad localization in motor neuron nuclei is strongly reduced in *gbb* nulls (Supplementary Fig. 4A, B, E). And in accordance with our model, while expression of Gbb in the muscle weakly rescues pMad localization in *gbb* nulls, neuronal Gbb provides significantly greater rescue (Supplementary Fig. 4A–E). These results argue that autocrine Gbb signaling is the primary BMP pathway at the L3 stage.

**Loss of Gbb phenocopies loss of α₂δ-3 at embryonic NMJs.** Our findings argue that Gbb regulates synapse maintenance; however, they do not address whether it regulates initial synapse formation. At 21 h after egg laying (AEL), embryonic NMJs display hallmarks of mature NMJs such as rounded boutons and localized presynaptic components, including Brp[43,44]. To address whether Gbb is required for embryonic synapse formation, we assessed Brp localization in *gbb* mutant embryos.

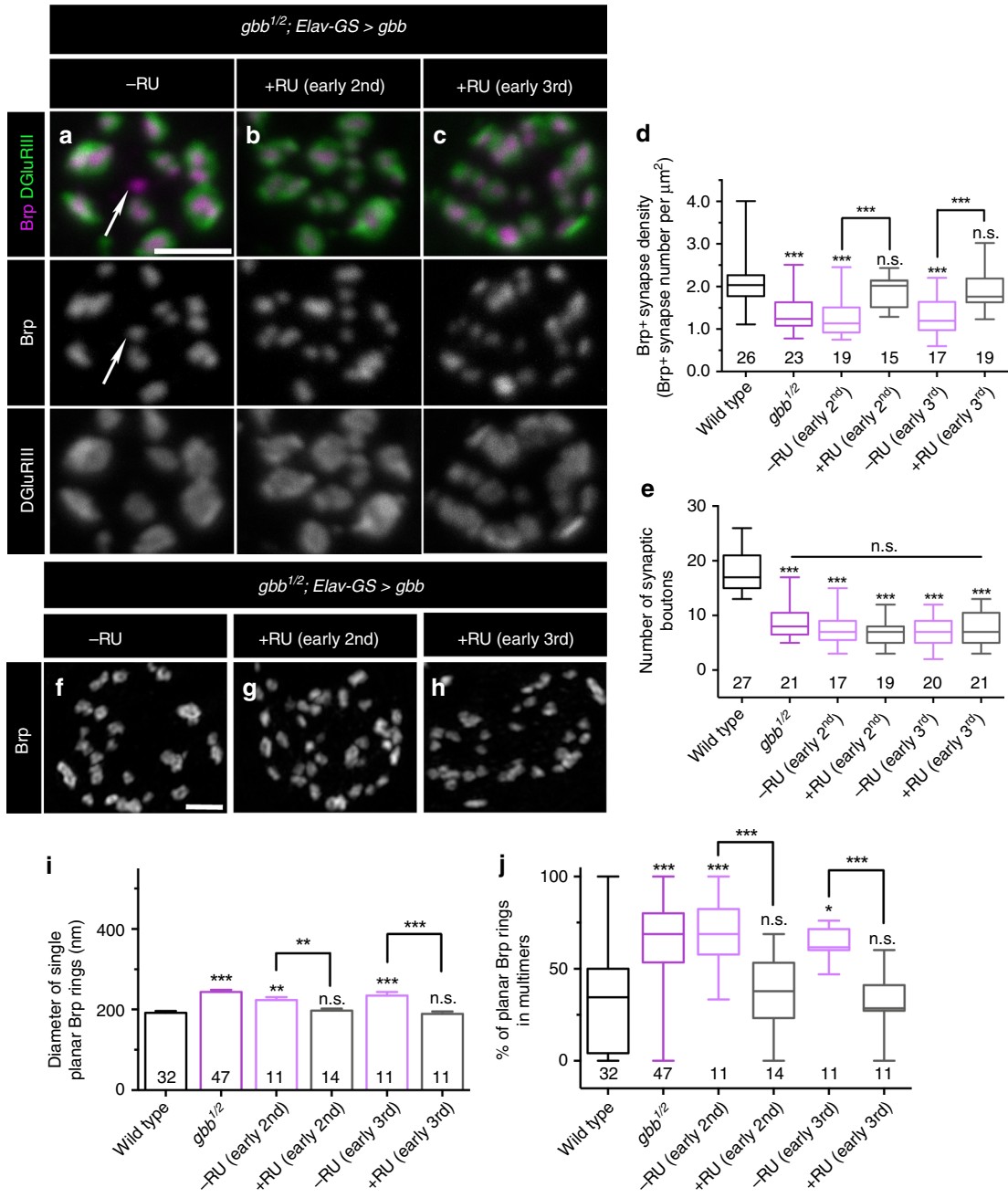

**Fig. 5 An ongoing requirement for presynaptic BMP signaling. a–c** Representative *z*-projections of boutons labeled with labeled with Brp (magenta) and DGluRIII (green) of *gbb*$^{1/2}$ mutants with transgenic *UAS-gbb* under the control of *ElavGeneSwitch*. Arrows denote unapposed Brp+puncta. Scale bar: 2 μm. **d** Quantification of Brp+synapse density. *n* is the number of boutons. **e** Quantification of the number of boutons. *n* is the number of NMJs scored. **f–h** Representative *z*-projections of boutons of the indicated genotypes and treatments labeled with anti-Brp. Scale bar: 1 μm. **i** Quantification of single planar Brp ring diameters. **j** Quantification of the percentage of planar Brp rings in multimers. For Brp ring analyses, *n* is the number of boutons scored. For the bar graph, error bars are mean ± SEM. For all box-and-whisker plots, error bars are min and max data points, and the center line indicates the median. n.s. not significantly different. *$p < 0.05$; **$p < 0.01$; ***$p < 0.001$. All tests are nonparametric Kruskal–Wallis one-way ANOVAs on ranks followed by Dunn's multiple comparison test.

We find approximately 40 Brp-positive puncta at NMJ 6/7 in wild type (Fig. 6a, f). Unexpectedly, while Brp-positive puncta number is unchanged in *gbb* or *wit* null embryos (Fig. 6a–c, f), the gross morphology of these NMJs is abnormal. Whereas wild-type embryonic NMJs are characterized by clusters of well-defined boutons, NMJs in *gbb* and *wit* mutants are elongated with fewer and smaller boutons. Specifically, we find a 50% reduction in bouton number in *gbb* and *wit* mutants (Fig. 6a–c, g). The motor neuron and muscle-specific drivers

used in our larval analysis did not permit us to map the cellular requirement of Gbb in embryos. As an alternative we assayed *crimpy* mutants. Crimpy is a neuronal Gbb-binding protein that traffics presynaptic Gbb to dense core vesicles (DCVs) for activity-dependent release[24]. Loss of Crimpy impairs embryonic bouton formation (Fig. 6d, g), raising the surprising possibility that while NMJ expansion depends on postsynaptic Gbb, initial bouton formation may be regulated by the presynaptic ligand pool.

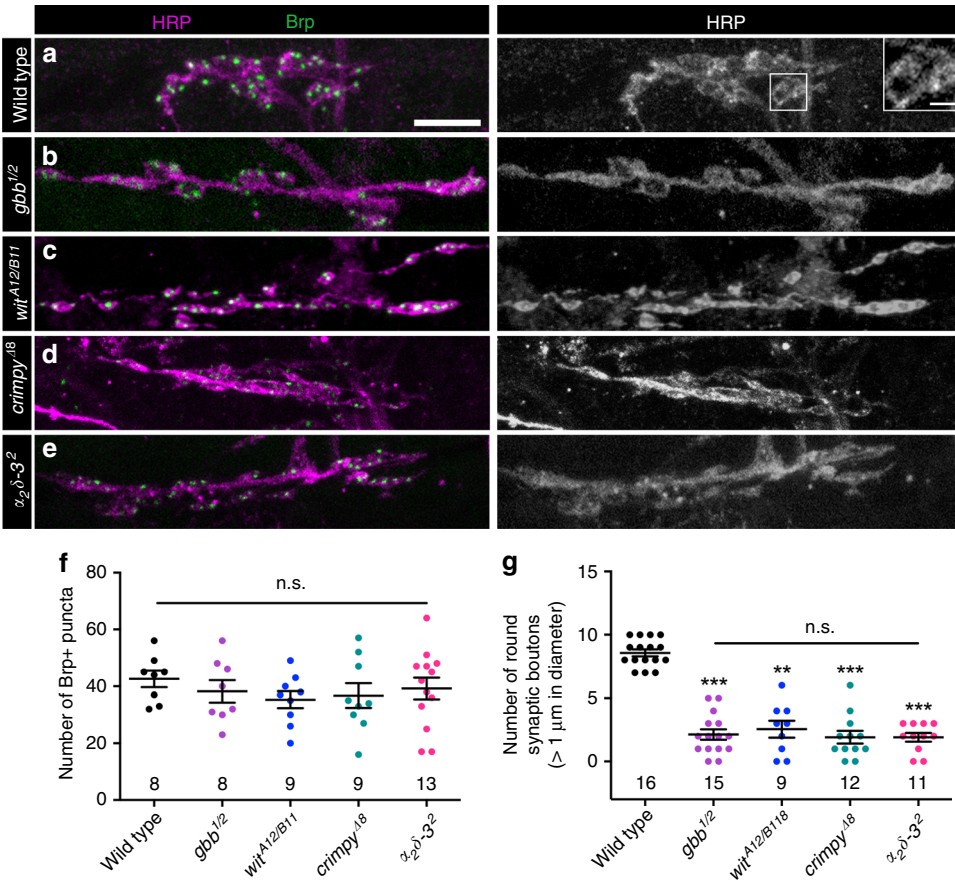

**Fig. 6 Loss of Gbb phenocopies loss of $\alpha_2\delta$-3 at embryonic NMJs. a–e** Representative *z*-projections of embryonic NMJs of the indicated genotypes labeled with Brp (green) and HRP (magenta). Scale bar: 2 μm. Inset in **a** depicts two synaptic boutons larger than 1 μm in diameter. Scale bar: 1 μm. **f** Quantification of the number of Brp+puncta. **g** Quantification of the number of synaptic boutons greater than 1 μm in diameter. For all embryonic experiments, *n* is the number of NMJs scored. Error bars are mean ± SEM. Individual data points are displayed as dots. n.s. not significantly different. **p* < 0.01; ***p* < 0.001. All tests are nonparametric Kruskal–Wallis one-way ANOVAs on ranks followed by Dunn's multiple comparison test.

This embryonic phenotype caught our attention because it mirrors the phenotype displayed by strong $\alpha_2\delta$-3 alleles (Fig. 6e–g)[17]. $\alpha_2\delta$-3 is an auxiliary $Ca^{2+}$ channel subunit that traffics and localizes Cacophany (Cac), the Drosophila $\alpha_1$ subunit of mammalian N- and P/Q-type voltage-gated calcium channels[18]. $\alpha_2\delta$-3 proteins contain two disulfide-linked peptides, $\alpha_2$ and $\delta$. While the C-terminal $\delta$ peptide is membrane-associated and necessary for $\alpha_1$ subunit association, the N-terminal $\alpha_2$ domain is exclusively extracellular[8]. $\alpha_2\delta$ proteins are conserved regulators of synapse formation and function[8,15,20,45–48]. Yet how they carry out their distinct functions in $Ca^{2+}$ channel localization versus synapse formation is not well understood.

**Gbb and $\alpha_2\delta$-3 have related functions in NMJ organization.** The phenotypic similarity between BMP mutants and $\alpha_2\delta$-3 mutants at embryonic NMJs raised the possibility of an underlying functional interaction. Hence, we tested if loss of $\alpha_2\delta$-3 results in presynaptic deficits, making use of well-characterized $\alpha_2\delta$-3 alleles. $\alpha_2\delta$-$3^{DD106}$ is a null allele; $\alpha_2\delta$-$3^{DD196}$ is a hypomorphic allele harboring a stop codon following the $\alpha_2$ peptide; and $\alpha_2\delta$-$3^{k10814}$ (here called $\alpha_2\delta$-$3^k$) is a hypomorphic P element allele with reduced expression of wild-type protein[17,18] (Fig. 7e). While bouton formation is blocked in $\alpha_2\delta$-$3^{DD106}$ homozygotes, $\alpha_2\delta$-$3^{DD196}$ mutants have normal embryonic NMJs, suggesting a specific role for the extracellular $\alpha_2$ peptide in bouton morphogenesis[17]. $\alpha_2\delta$-$3^{DD106}$ homozygotes are late-stage embryonic

lethal, so to characterize presynaptic organization at the L3 stage, we analyzed this allele in trans to the weak $\alpha_2\delta$-$3^k$ allele, as well as $\alpha_2\delta$-$3^{DD196}$ in trans to $\alpha_2\delta$-$3^k$. Consistent with $\alpha_2\delta$-3's role in $Ca^{2+}$ channel localization, both allelic combinations ($\alpha_2\delta$-$3^{DD106/k}$ and $\alpha_2\delta$-$3^{DD196/k}$) display striking reductions of Cac at active zones (Supplementary Fig. 5A–C, I)[49].

We next assessed presynaptic organization in $\alpha_2\delta$-$3^{DD106/k}$ and $\alpha_2\delta$-$3^{DD196/k}$ animals. We find that $\alpha_2\delta$-$3^{DD106/k}$ mutant NMJs have synaptic defects analogous to those observed in presynaptic Gbb mutants. Specifically, the density of Brp/GluRIII pairs decreases (Fig. 7a–d), while the diameter of isolated planar Brp rings and the percentage of interconnected Brp rings increases (Fig. 7f–j). The observed decrease in synapse density in $\alpha_2\delta$-$3^{DD106/k}$ mutants is similar to that reported for another strong $\alpha_2\delta$-3 allelic combination[18]. $\alpha_2\delta$-$3^{DD106/k}$ mutants also display altered SSV distribution akin to presynaptic Gbb mutants (Fig. 7k–o). In contrast, presynaptic organization appears normal in $\alpha_2\delta$-$3^{DD196/k}$ mutants (Fig. 7a–o). These findings indicate that $\alpha_2\delta$-3 regulates active zone cytomatrix architecture and SSV distribution and suggest a selective requirement for the extracellular $\alpha_2$ domain. We went on to assay whether BMP signaling is required for Cac localization, an established function of $\alpha_2\delta$-3. While Cac is almost perfectly co-localized with Brp in controls, approximately 20% of Brp puncta lack Cac in BMP pathway mutants (Supplementary Fig. S5A, D–F, I). We wondered if impaired Cac localization might be a general phenotype of synapse-organizing mutants, so investigated it in *nrx-1*$^{241/273}$ and

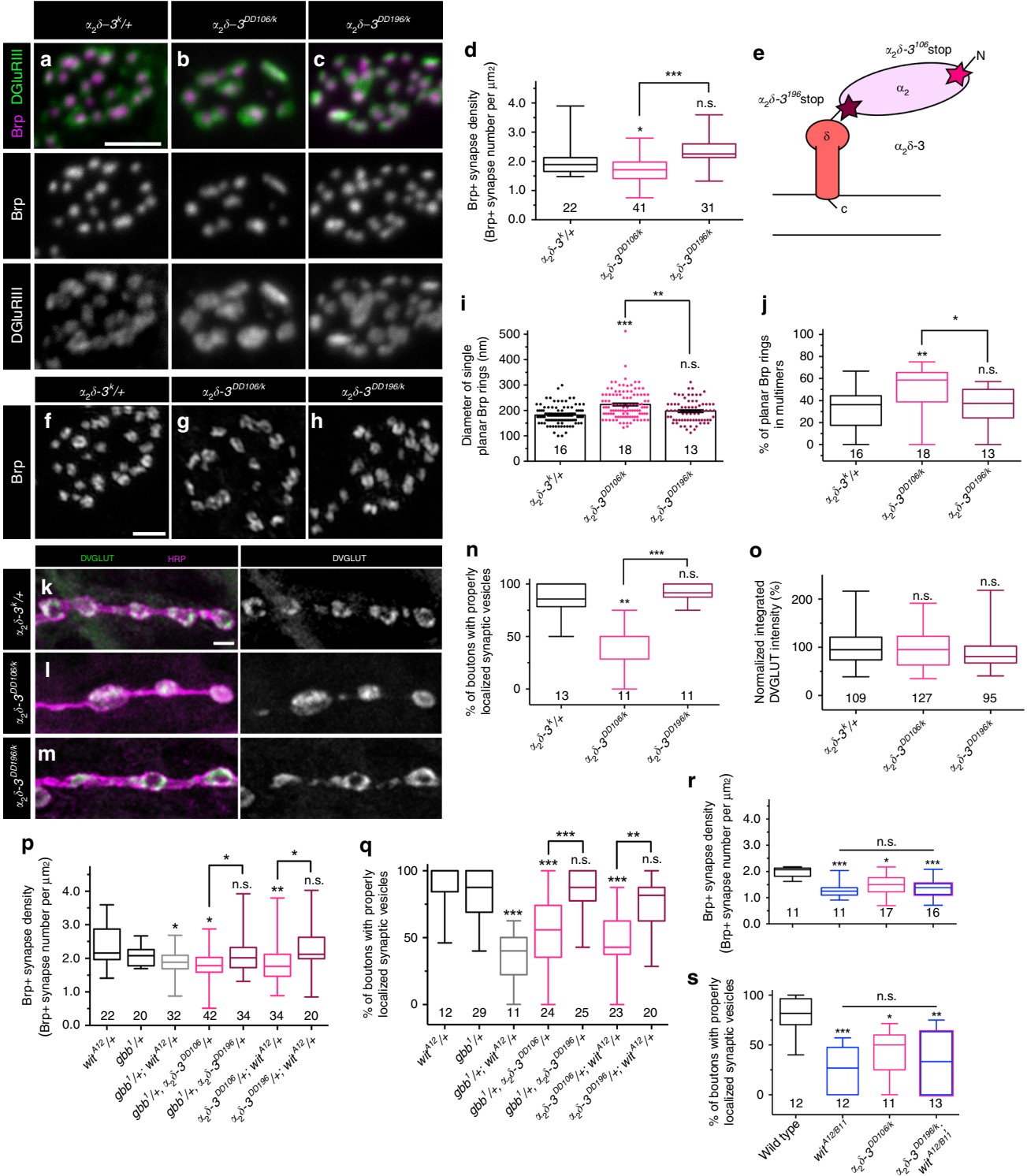

<em>syd-1</em>$^{ex1.2/ex3.4}$ mutants. However, Cac localization to Brp-positive active zones is normal in both backgrounds (Supplementary Fig. 5G–I). These data suggest a relatively more selective requirement for BMP signaling in Cac localization and define broad phenotypic similarities between $\alpha_2\delta$-3 and BMP pathway mutants.

We next assayed for genetic interactions between BMP pathway mutants and $\alpha_2\delta$-3 mutants. In line with a common pathway, $\alpha_2\delta$-3$^{DD106}$, <em>gbb</em>$^1$ and $\alpha_2\delta$-3$^{DD106}$; <em>wit</em>$^{A12}$ double heterozygotes display decreased Brp-positive synapse density

(Fig. 7p) and aberrant SSV distribution (Fig. 7q) similar to individual single mutants. Notably, the strength of the interaction between $\alpha_2\delta$-3$^{DD106}$ and <em>gbb</em> or <em>wit</em> is as strong as the genetic interaction between <em>gbb</em> and <em>wit</em> (Fig. 7p, q). We next analyzed the phenotype of $\alpha_2\delta$-3$^{DD106/k}$, <em>wit</em>$^{A12/B11}$ double homozygotes. We find that the phenotypes of $\alpha_2\delta$-3$^{DD106/k}$, <em>wit</em>$^{A12/B11}$ double mutants are no more severe than either individual single mutant (Fig. 7r, s). Together, these genetic interactions suggest that $\alpha_2\delta$-3 functions in the autocrine BMP signaling pathway.

**Fig. 7 Gbb and $\alpha_2\delta$-3 have related functions in presynaptic organization. a–c** Representative *z*-projections of boutons of the indicated genotypes labeled with Brp (magenta) and DGluRIII (green). Scale bar: 2 μm. **d** Quantification of the Brp+ synapse density. *n* is the number of boutons scored. **e** Schematic of $\alpha_2\delta$-3 subunit of a voltage-dependent Ca$^{2+}$ channel. Stars mark the locations of the stop codons in $\alpha_2\delta$-3$^{DD106}$ and $\alpha_2\delta$-3$^{DD196}$ alleles. **f–h** Representative deconvolved *z*-projections of boutons of the indicated genotypes labeled with anti-Brp. Scale bar: 1 μm. **i** Quantification of single planar Brp ring diameters. **j** Quantification of the percentage of planar Brp rings in multimers in boutons. For Brp ring analyses, *n* is the number of boutons scored. **k–m** Representative *z*-projections of boutons of the indicated genotypes labeled with DVGLUT (green) and HRP (magenta). Scale bar: 2 μm. **n** Quantification of the percentage of boutons exhibiting properly localized synaptic vesicles. *n* is the number of NMJs scored. **o** Quantification of integrated DVGLUT intensity compared to proper controls. *n* is the number of boutons scored. **p** Quantification of Brp+ synapse density. **q** Quantification of the percentage of boutons exhibiting properly localized synaptic vesicles. **r** Quantification of Brp+ synapse density. *n* is the number of boutons scored. **s** Quantification of the percentage of boutons exhibiting properly localized synaptic vesicles. *n* is the number of NMJs scored. For the bar graph, error bars are mean ± SEM. For all box-and-whisker plots, error bars are min and max data points, and the center line indicates the median. Individual data points are displayed as dots. n.s. not significantly different. *$p < 0.05$; **$p < 0.01$; ***$p < 0.001$. All tests are nonparametric Kruskal–Wallis one-way ANOVAs on ranks followed by Dunn's multiple comparison test.

To judge the specificity of these genetic interactions, we tested if either *gbb* or $\alpha_2\delta$-3 displays dominant genetic interactions with *nrx-1* or *syd-1*. We find that all four double heterozygote combinations (*gbb*$^1$/+; *Nrx-1*$^{241}$/+, *gbb*$^1$/+; *Syd-1*$^{ex1.2}$/+, $\alpha_2\delta$-3$^{106}$/+; *Nrx-1*$^{241}$/+, and $\alpha_2\delta$-3$^{106}$/+; *Syd-1*$^{ex1.2}$/+) display normal Brp-positive synapse density (Supplementary Fig. 5J) and normal SSV distribution (Supplementary Fig. 5K). Thus, the observed genetic interactions between $\alpha_2\delta$-3 and the BMP pathway are not a generic effect of heterozygosity for two synapse organizers and suggest a more intimate functional relationship between $\alpha_2\delta$-3 and BMP signaling.

**Overexpressing presynaptic Gbb suppresses $\alpha_2\delta$-3 phenotypes.** We considered two hypotheses for how $\alpha_2\delta$-3 and BMP signaling may interact. First, $\alpha_2\delta$-3 might be required for autocrine BMP signaling, perhaps by serving as an obligate co-receptor. Alternatively, $\alpha_2\delta$-3 might act in the extracellular space to potentiate autocrine BMP signaling by regulating Gbb levels or diffusion. To distinguish between these hypotheses, we tested if elevating autocrine BMP signaling suppresses $\alpha_2\delta$-3 phenotypes. We reasoned that if $\alpha_2\delta$-3 facilitates autocrine BMP pathway activity, then increasing presynaptic Gbb might suppress $\alpha_2\delta$-3 mutant phenotypes. In contrast, if $\alpha_2\delta$-3 is an essential BMP pathway component, then elevated Gbb is not predicted to alter $\alpha_2\delta$-3 mutant phenotypes. We find that the decrease in Brp-positive synapse density observed in $\alpha_2\delta$-3$^{DD106/k}$ animals is suppressed by overexpressing Gbb presynaptically (Fig. 8a–d). Similarly, Brp ring size/distribution and proper SSV distribution is restored to $\alpha_2\delta$-3 mutants by neuronal overexpression of Gbb (Fig. 8e–m). These findings suggest $\alpha_2\delta$-3 normally facilitates autocrine BMP signaling, but may not be absolutely essential for its function. To probe the specificity of the relationship between the presynaptic ligand and $\alpha_2\delta$-3, we tested if postsynaptic Gbb overexpression likewise suppresses $\alpha_2\delta$-3 mutant phenotypes. However, Gbb overexpression in muscle has no effect on the size or density of Brp puncta (Supplementary Fig. 6A–I) and does not reverse SSV de-localization in $\alpha_2\delta$-3 mutants (Supplementary Fig. 6J–M). These results suggest a selective deficit in autocrine BMP signaling in $\alpha_2\delta$-3 mutants.

In the course of working with the $\alpha_2\delta$-3 mutants, we noticed that they were markedly sluggish and wondered if presynaptic overexpression of Gbb improves motor function of $\alpha_2\delta$-3 mutant animals. Thus, we compared larval locomotion of $\alpha_2\delta$-3 mutants and $\alpha_2\delta$-3 mutants with neuronal Gbb overexpression. We find that $\alpha_2\delta$-3$^{DD106/k}$ mutants exhibit dramatically reduced larval locomotion relative to wild type, moving only 18% of the distance as wild-type larvae over a 3-min interval (Fig. 8n, o, q). Motor neuron-specific overexpression of Gbb results in a twofold increase in the distance traveled by $\alpha_2\delta$-3$^{DD106/k}$ animals

(Fig. 8n–q). Given the separate requirement for $\alpha_2\delta$-3 in calcium channel localization, it is striking that elevated autocrine BMP signaling improves larval locomotion of $\alpha_2\delta$-3 animals. Together, these data support the hypothesis that $\alpha_2\delta$-3 facilitates BMP signaling.

**$\alpha_2\delta$-3 limits diffusion of presynaptic Gbb.** The close functional relationship between $\alpha_2\delta$-3 and presynaptic Gbb raises the possibility of a physical interaction. To explore this possibility, we first assessed $\alpha_2\delta$-3 localization. We tested if $\alpha_2\delta$-3 is detectable at NMJs with an antibody used to characterize $\alpha_2\delta$-3 expression in the fly brain[50]. We find robust expression of $\alpha_2\delta$-3 at control NMJs (Fig. 9a), which is reduced in $\alpha_2\delta$-3$^{DD106/k}$ hypomorphs (Fig. 9a, c). To investigate if $\alpha_2\delta$-3 is in a position to interact with Gbb following its activity-dependent release, we expressed Gbb-HA in motor neurons and drove Gbb release by stimulating in high K$^+$[24]. We find substantial co-localization of $\alpha_2\delta$-3 and Gbb, particularly along the exterior surface of the bouton (Fig. 9d).

To test whether $\alpha_2\delta$-3 and Gbb are close enough to interact, we utilized a proximity ligation assay (PLA). In this technique, primary antibodies are recognized with secondary antibodies coupled to oligonucleotide probes, which generate a fluorescent signal only when the epitopes are within 40 nm of each other[51,52]. We expressed Gbb-HA in motor neurons, stimulated its release in high K$^+$, and asked if a signal is detected when PLA secondaries recognizing both Gbb-HA and $\alpha_2\delta$-3 are added. If the probes are not added, no fluorescent PLA signal is detected (Supplementary Fig. 7A). However, strong positive signal is detected in the presence of both probes (Fig. 9e), indicating that $\alpha_2\delta$-3 and Gbb are in close proximity to each other, suggestive of a physical interaction.

Extracellular proteins serve many functions in BMP pathways. They can antagonize BMP signaling by sequestering BMP ligands from their receptors, or alternatively, promote BMP signaling by serving as scaffolds to facilitate ligand–receptor complex formation[53]. We hypothesized that the $\alpha_2$ peptide is well-positioned to modulate the extracellular distribution of Gbb. In this case, we might observe a shift in extracellular Gbb in the presence and absence of $\alpha_2\delta$-3. To test this prediction, we stimulated Gbb release and quantified it by comparing Gbb-HA intensity within the presynaptic membrane in unstimulated and stimulated preparations. Following stimulation, we find a decrease in mean Gbb-HA intensity within the presynaptic membrane (Fig. 10a, b, g)[24]. We next investigated if loss of $\alpha_2\delta$-3 interferes with presynaptic Gbb localization or release. We do not find alterations in presynaptic Gbb-HA levels in the absence of stimulation in either $\alpha_2\delta$-3 mutant background, and moreover, stimulation drives comparable reductions of intracellular Gbb-HA in both backgrounds (Fig. 10a–g). These findings are

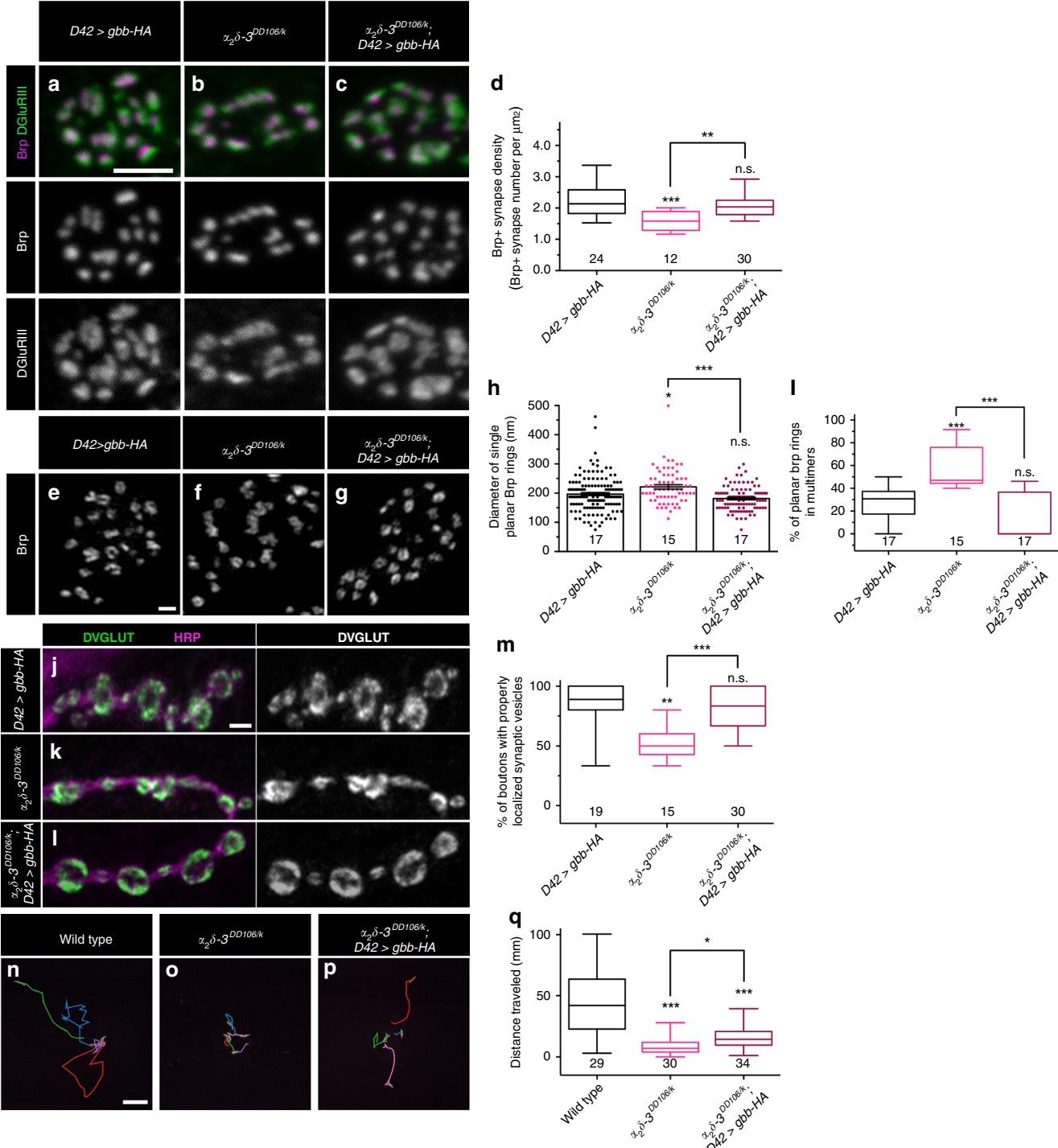

**Fig. 8 Overexpressing presynaptic Gbb suppresses $\alpha_2\delta$-3 mutant phenotypes. a–c** Representative z-projections of boutons of the indicated genotypes labeled with Brp (magenta) and DGluRIII (green). Scale bar: 2 μm. **d** Quantification of the Brp + synapse density. n is the number of boutons scored. **e–g** Representative deconvolved z-projections of boutons of the indicated genotypes labeled with anti-Brp. Scale bar: 1 μm. **h** Quantification of single planar Brp ring diameters. (**i**) Quantification of the percentage of planar Brp rings in multimers in boutons. For Brp ring analyses, n is the number of boutons scored. **j–l** Representative z-projections of boutons of the indicated genotypes labeled with DVGLUT (green) and HRP (magenta). Scale bar: 2 μm. **m** Quantification of the percentage of boutons exhibiting properly localized synaptic vesicles. n is the number of NMJs scored. **n–p** Representative traces of third-instar larvae of the indicated genotypes crawling for 3 min. Each color represents an individual larva. Scale bar: 10 mm. **q** Quantification of the total distance traveled by larvae of the indicated genotypes. n is the number of larvae scored. For the bar graph, error bars are mean ± SEM. For all box-and-whisker plots, error bars are min and max data points, and the center line indicates the median. Individual data points are displayed as dots. n.s., not significantly different. *$p < 0.05$; **$p < 0.01$; ***$p < 0.001$. All tests are nonparametric Kruskal-Wallis one-way ANOVAs on ranks followed by Dunn's multiple comparison test.

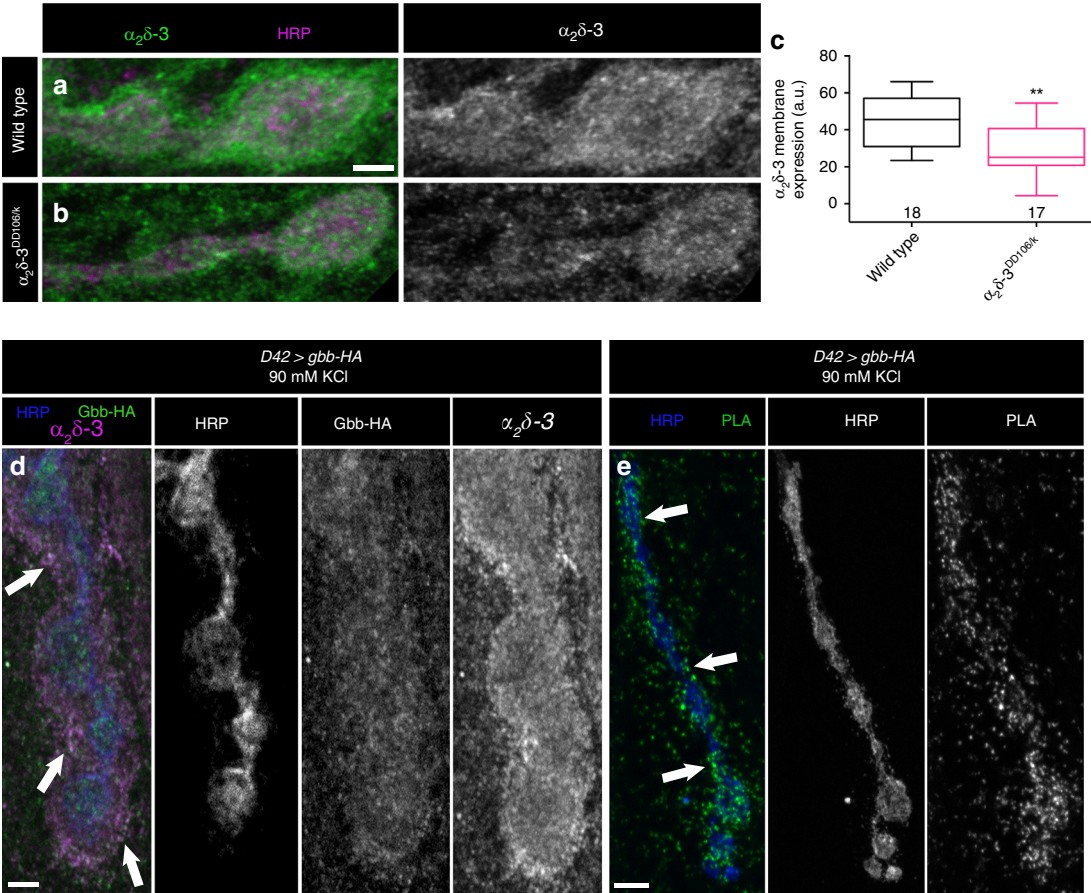

**Fig. 9 $\alpha_2\delta$-3 and presynaptic Gbb are in close proximity. a, b** Representative *z*-projections of boutons of the indicated genotypes labeled with HRP (magenta) and $\alpha_2\delta$-3 (green). Scale bar: 2 μm. **c** Quantification of $\alpha_2\delta$-3 membrane expression. *n* is the number of NMJs scored. \*\**p* < 0.01 by a two-tailed Mann–Whitney test. **d** Representative *z*-projections of boutons of the indicated genotype after neuronal stimulation labeled with HRP (blue), $\alpha_2\delta$-3 (magenta), and Gbb-HA (green). Arrows denote areas of co-localization of presynaptic Gbb and $\alpha_2\delta$-3. Scale bar: 2 μm. **e** Representative *z*-projection of boutons labeled with HRP (blue) and PLA-specific probes (green) to detect proximity ligation events. Arrows denote close proximity of presynaptic Gbb and $\alpha_2\delta$-3 as exhibited by expression of PLA-specific probes along the rim of the neuronal membrane. Scale bar: 5 μm.

consistent with the model that $\alpha_2\delta$-3 is an extracellular modulator of autocrine BMP signaling.

We then assessed if $\alpha_2\delta$-3 acts subsequently to shape the distribution of Gbb following its activity-dependent release. At control NMJs, released Gbb-HA is visible as a cloud around the presynaptic membrane following stimulation (Fig. 10a (top), b (bottom)). We quantified extracellular Gbb by defining a 400-nm margin from the intracellular face of the presynaptic membrane as labeled by HRP into the extracellular space (between dashed lines in Fig. 10a–f (middle)). At control NMJs, we find a twofold increase in mean Gbb-HA intensity in this membrane-proximal domain following stimulation (Fig. 10a (top), b (bottom), h)[24]. In contrast, we do not detect an increase in membrane-proximal Gbb following stimulation at $\alpha_2\delta$-3$^{DD106/k}$ mutant NMJs (Fig. 10c (top), d (bottom), h). To address a specific function for the extracellular $\alpha_2$ peptide, we analyzed the distribution of Gbb-HA following stimulation in $\alpha_2\delta$-3$^{DD196/k}$ mutants. Unlike $\alpha_2\delta$-3$^{DD106/k}$ mutants, the extracellular distribution of Gbb-HA following stimulation is normal at $\alpha_2\delta$-3$^{DD196/k}$ mutant NMJs (Fig. 10e (top), f (bottom), h). We repeated this analysis, tightening the margin to 200 nm, and observed the same result; namely, a significant increase in Gbb-HA in this region in controls and $\alpha_2\delta$-3$^{DD196/k}$ mutant NMJs, but not in $\alpha_2\delta$-3$^{DD106/k}$ mutant NMJs (Supplementary Fig. 8A). These findings argue that $\alpha_2\delta$-3 may provide a physical barrier to Gbb diffusion to promote

activity-dependent autocrine BMP signaling and further support a specific function for the extracellular $\alpha_2$ peptide in the pathway.

## Discussion

In this study we uncovered a functional interaction between $\alpha_2\delta$-3 and an activity-dependent, autocrine BMP signaling pathway at the Drosophila NMJ (Fig. 10i). BMP pathways regulate bouton number and morphology, bouton stabilization, presynaptic organization, transmitter release, activity-dependent bouton addition, homeostatic plasticity, and postsynaptic differentiation at the Drosophila NMJ (this work)[21,24–27,38,54–58]. Multiple lines of evidence indicate that these pathways are at least partially separable. For instance, both bouton stabilization and activity-dependent bouton addition are non-canonical as they activate LIM kinase directly downstream of the Wit receptor[55,56]. A second non-canonical BMP pathway regulates glutamate receptor composition at the NMJ. Interestingly, it requires local, presynaptic accumulation of pMad[54,58]. While accumulation of synaptic pMad depends on Wit, it is independent of Gbb, indicating that it is separable from the Gbb-dependent autocrine pathway described in this study.

How are the Mad-dependent pathways distinguished? In other words, how does retrograde BMP signaling regulate Mad-dependent gene transcription to promote scaling growth while

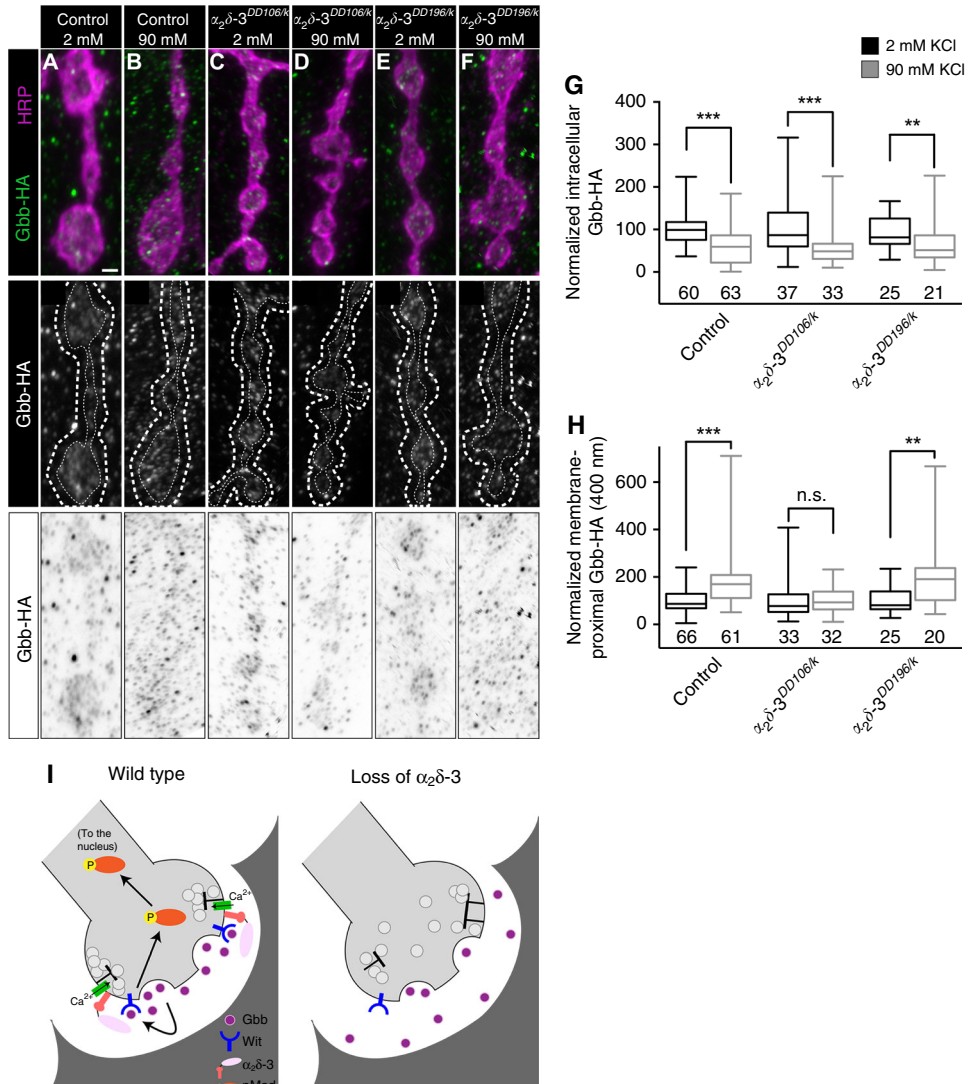

**Fig. 10 α₂δ-3 limits diffusion of Gbb following its activity-dependent release. a–f** Top: representative z-projections of boutons of the indicated genotypes labeled with HRP (magenta) and Gbb-HA (green). Genotypes specifically are: control *(D42>gbb-HA)*, α₂δ-3^DD106/k *(α₂δ-3^DD106/k10814; D42>gbb-HA)*, and α₂δ-3^DD196/k *(α₂δ-3^DD196/k10814; D42>gbb-HA)*. Scale bar: 1 μm. Middle: individual Gbb-HA channel shown in grayscale. Membrane-proximal Gbb-HA was measured between the two dashed white lines. Bottom: individual Gbb-HA channel with inverted colors. **g** Quantification of intracellular Gbb-HA normalized to control levels before (2 mM KCl) and after (90 mM KCl) neuronal stimulation. For all Gbb-HA release experiments, *n* is the number of NMJs scored. **h** Quantification of membrane-proximal Gbb-HA (within 400 nm) normalized to control levels before and after neuronal stimulation. n.s. not significantly different. **p < 0.01; ***p < 0.001 by a two-tailed Mann–Whitney test. **i** Proposed model for α₂δ-3 acting as a physical barrier to promote Gbb signaling. Upon release from the neuron, Gbb is released into the synaptic cleft. With the aid of α₂δ-3, Gbb remains in close proximity to the presynaptic membrane and is then able to activate the BMP Type II receptor Wit. The receptor complex in turn phosphorylates the transcription factor Mad, transducing a BMP signal back to the nucleus of the neuron.

autocrine BMP signaling regulates Mad-dependent gene transcription to maintain synapse structure and function? The distinct temporal requirement for the two pathways likely plays a key role to enable signal segregation. While the L1 stage is the critical period for the retrograde pro-growth signal[22], we show that ongoing autocrine BMP signaling maintains synaptic structure. Thus, the signals are largely temporally separable. However, sustained muscle overexpression of Gbb during larval development fails to rescue baseline glutamate release indicating that temporal differences alone do not explain the distinct functions of the two pathways[21,26].

We propose that the temporal dynamics of activity-dependent release of BMP from presynaptic terminals promote robust induction of the downstream BMP cascade. This hypothesis is consistent with work demonstrating that pulsed TGF-β signaling dramatically increases the Smad transcriptional response[59,60]. This model is also supported by our work on neuronal Gbb trafficking and release. Neuronal Gbb is not constitutively released, but rather is delivered to DCVs and is released in response to neuronal activity[24]. In the absence of Cmpy-mediated Gbb release, the ability of presynaptic Gbb to promote baseline neurotransmission is impaired. Interestingly, Cmpy contains an IGF-binding domain and is likely to modulate the function of additional growth factors. Thus, here we have focused on Gbb.

The signaling range of BMP ligands in the extracellular space is tightly controlled by interacting proteins, including ECM components[53,61,62]. Not surprisingly, the ECM serves context-dependent functions in BMP signaling. For example, fibrillins

sequester BMPs in the ECM to inhibit signaling[63,64], while collagen type IV interacts with BMPs to augment local BMP signaling[65–67]. More broadly, the ECM is proposed to generate a structural barrier through which BMP diffusion is limited[53]. Such a barrier function could reasonably play a critical activating role in autocrine BMP pathways, such as the one described here.

Is $\alpha_2\delta$-3's only function in the synaptic cleft local modulation of autocrine BMP signaling? Mammalian $\alpha_2\delta$-1 binds a large number of synaptic proteins in addition to thrombospondin, including integrin subunits, neurexin, neuroligin, and cadherins[68], suggesting that $\alpha_2\delta$ proteins interact widely with synaptic signaling or adhesion complexes. While we found clear phenotypic similarities between autocrine BMP mutants and $\alpha_2\delta$-3 mutants, we could not analyze strong allelic combinations of $\alpha_2\delta$-3 in late larval development as its function in Cac localization renders strong alleles late-embryonic lethal. To determine if $\alpha_2\delta$-3 modulates other extracellular signals, it would be informative to generate $\alpha_2\delta$-3 alleles that specifically delete the $\alpha_2$ domain to distinguish $\alpha_2\delta$-3's function in Cac localization from its Cac-independent roles in modulating extracellular signaling or adhesion.

The autocrine BMP pathway we describe here is predicted to be transcriptional since it requires Mad. It will be essential to identify the cohort of Mad transcriptional targets induced by this activity-dependent cascade. Intriguingly, BMP4 was recently found to be a presynaptic, autocrine cue in the mammalian hippocampus that signals through a canonical Smad pathway[69]. And similar to presynaptic Gbb in Drosophila, BMP4 is trafficked to DCVs and released in response to activity. However, in contrast to the synaptic maintenance function we describe, BMP4 signaling destabilizes synapses for synaptic elimination. Thus, activity-dependent autocrine BMP signaling plays a conserved role in regulating synapse density during development, though whether the pathway has a stabilizing or destabilizing effect on synapses is context-dependent. It will be essential to determine the molecular mechanisms through which activity-dependent BMP signal transduction exerts its broad range of effects on synapse structure and function.

## Methods

**Fly stocks.** The following stocks were used: *OregonR* (wild type), *BG57Gal4*, *D42Gal4*, *Elav-GeneSwitch*[42], *gbb1* and *UAS-gbb9.1* (ref. [70]), *gbb2*, *UAS-gbb-RNAi*[71], *wit[A12]*, *wit[B11]*, and *UAS-wit27*[27], *mad12* (ref. [72]), *mad1* (ref. [73]), *crimpy[Δ8]* (ref. [23]), *cac[sfGFP-N]*[49], *α₂δ-3²* (ref. [19]), *α₂δ-3[DD106]*, *α₂δ-3[DD196]*, and *α₂δ-3[k10814]* (ref. [18]), *UAS-gbb-1xHA*[24], *syd-1[ex1.2]*, and *syd-1[ex3.4]* (ref. [32]), *nrx273* and *nrx241* (ref. [40]). All experiments were performed at 25 °C and in compliance with all relevant ethical regulations for animal testing and research at CWRU.

**Primers.** The primers used in this study are as follows: *wit[B11]*; F: CCACTCCGATTCCGAGCGAGCCAC, R: CCGTCCCTCATTGGAGCCCAGTTC, *UAS-gbb-1xHA*; F: CGCGAGGTGAAGCTGGACGACATT, R: GTGGTACAGAACGGGTAGTGCTCC, *D42Gal4*; F: CGATGATGAAGATACCCCACCAAACC, R: TAAACCGCTTGGAGCTTCGTCACG.

**Immunohistochemistry and immunofluorescence.** Wandering third-instar (L3) larvae were dissected in PBS and fixed in either Bouin's fixative for 5 min or 4% PFA in PTX (PBS and 0.1% Triton X-100) for 10 min (for Brp/GluRIII apposition) or 45 min (for Gbb-HA experiments). Larval fillets were then blocked in PBT (PBS, 1% BSA, and 0.1% Triton X-100) and incubated with primary antibody overnight at 4 °C. Samples were next washed in PBT and incubated with secondary antibody for 2 h at room temperature (RT). For Gbb-HA experiments, samples were incubated with primary antibody for 2 h at RT and overnight at 4 °C, and all washes and incubations were conducted in PBS. All samples were finally washed in PTX and mounted in either ProLong Diamond (Thermo Fisher Scientific) for Brp ring experiments or 60% glycerol for all other experiments.

For embryonic experiments, late-stage embryos were transferred using a small paintbrush to piece of double-sided tape. The embryos were rolled out of their chorion and placed on a plate in a small drop of PBS. Late-stage embryos have trachea fully formed and were removed from their vitelline membrane with forceps. Embryos were pinned at the head and tail using sharpened pins. Tungsten wire was used to pierce the body wall, and scissors were used to make a lateral cut down the stretched embryo. PBS was replaced with Bouin's fixative for 5 min. The embryo was then rinsed with PBS and placed into a mesh basket surrounded by PBS in a small Petri dish. The embryos in the basket were washed three times for 15 min in PBT while slowly shaking. Embryos were blocked with PBT with 10% NGS (PBTN) for 10–30 min and incubated with primary antibody diluted in PBTN overnight at 4 °C. Samples were then washed in PBT for 45 min while shaking, blocked in PBTN for 10–30 min, and incubated with secondary antibody diluted in PBTN for 2 h at RT. Embryos were washed in PBT three times for 15 min each and PBS twice for 5 min each prior to being mounted in 60% glycerol.

For larval CNS experiments, wandering third-instar larvae were dissected in Schneider's media (Sigma) supplemented with 10% FBS (Sigma) and 10 U/mL Penicillin–Streptomycin (Thermo Fisher Scientific), then fixed in 4% PFA for 20 min. They were transferred to borosilicate tubes and washed in PBS and blocked in PTN (PBS, 0.1% Triton X-100, and 1% NGS) at RT on a rocker for at least 1 h prior to overnight incubation in primary antibodies diluted in PTN at 4 °C. Secondary antibody incubation in PTN was done for 3 h at RT after primary antibody was washed out with PBS. Samples were mounted in ProLong Gold Antifade with DAPI (Thermo Fisher Scientific) after washing in PBS.

The following primary antibodies were used: rabbit anti-Bruchpilot (Developmental Studies Hybridoma Bank, NC82) at 1:100, rabbit anti-GluRIII (gift from A. DiAntonio) at 1:2500, rabbit anti-DVGLUT (gift from A. DiAntonio) at 1:10,000, rabbit anti-GFP (Abcam, ab6556) at 1:1000, rabbit anti-pMad (Cell Signaling, 41D10) at 1:100, guinea pig anti-Eve (gift from J. Skeath) at 1:100, rat anti-HA (Roche, 11-867-423-001) at 1:200, guinea pig anti-α₂δ-3 (ref. [50]) at 1:500, Alexa Fluor 647-conjugated HRP (Jackson ImmunoResearch) at 1:300, and DyLight-594 anti-HRP (Jackson ImmunoResearch) at 1:300. The following goat species-specific secondary antibodies were used: Alexa Fluor 488 and Alexa Fluor 568 (Invitrogen) at 1:300.

**RU486-GeneSwitch.** Induction of Gal4-dependent transcription in GeneSwitch-Gal4 lines is induced by feeding animals with RU486 since Gal4 is rendered steroid-inducible via fusion with the human progesterone receptor. Per Osterwalder et al.[42], adults and their larval offspring were raised on non-RU486 food. L2 or early L3 larvae were transferred to molasses plates containing 25 mg/mL RU486 (mifepristone; Sigma) and topped with a yeast paste made from 1 g dried yeast and 2 mL 50 mg/mL RU486 in water. RU486 was prepared as a stock solution at 10 mg/mL in ethanol. Control larvae were fed food containing equivalent concentrations of ethanol as vehicle at the same developmental stage. We ensured that in the absence of RU486, *gbb[−/−]; Elav-GS>gbb* animals display active zone phenotypes similar to that observed in *gbb* nulls. Indeed, without drug, we saw identical structural phenotypes as observed in *gbb* nulls, confirming RU486-dependence of transgene expression.

**Gbb-HA release following neuronal stimulation.** Each larva was pinned at the head and tail while in cold hemolymph-like HL3 saline. An incision was made laterally, and one pin was moved closer to the other, so the body wall was relaxed. The HL3 was replaced with either cold 2 mM K⁺ (45 mM NaCl, 2 mM KCl, 4 mM MgCl₂, 36 mM sucrose, 5 mM HEPES, and 2 mM CaCl₂, pH 7.3) or 90 mM K⁺ (45 mM NaCl, 90 mM KCl, 4 mM MgCl₂, 36 mM sucrose, 5 mM HEPES, and 2 mM CaCl₂, pH 7.3) Jan's saline for 5 min. The larva was then quickly dissected in cold PBS and fixed.

**Proximity ligation assay.** We performed Duolink proximity ligation assay (PLA) on NMJ preparations. First, custom-made PLA probes were made using Duolink Probemaker Kits (Sigma, DUO92009 and DUO92010). Here, the PLUS oligo arm was conjugated to a goat anti-rat secondary antibody (Jackson ImmunoResearch, 712-005-150), and the MINUS oligo arm was conjugated to a goat anti-guinea pig secondary antibody (Jackson ImmunoResearch, 706-005-148).

*D42>gbb-HA* larvae were stimulated with 90 mM K⁺ solution per the Gbb-HA release protocol above. Larvae were then fixed in 4% PFA and incubated with rat anti-HA and guinea pig anti-α₂δ-3 for 2 h at RT and overnight at 4 °C with Alexa Fluor 647-conjugated HRP (Invitrogen) at 1:150. After washing in PBS three times for 5 min each, custom-made PLA probes (anti-rat PLUS and anti-guinea pig MINUS) diluted in the Probemaker Kit dilution buffer (1:5) were incubated for 1 h at 37 °C. Unbound PLA probes were removed by washing with Buffer A twice for 5 min each. Larvae were then incubated in the ligation solution comprised of the Duolink Ligation stock (1:5) and Duolink Ligase (1:40) for 30 min at 37 °C. After washing with Buffer A twice for 2 min each, the amplification step was conducted by incubating the larvae in the amplification stock (1:5) and the polymerase (1:80) for 1 h at 37 °C. Finally, larvae were washed in 1× Buffer B twice for 10 min each and 0.01× Buffer B for 1 min. Larvae were mounted in ProLong Gold Antifade (Thermo Fisher Scientific). The target proteins were visible in green (Sigma, DUO92014). Controls without probes went through an identical process, but water was substituted for the probes, ligation solution, and amplification solution.

**Image acquisition and quantification.** NMJs were imaged on a Zeiss LSM 800 confocal microscope with ×100 1.4 NA and ×40 1.3 NA oil-immersion objectives. Complete z-stacks were acquired with optimized settings to ensure that over-saturation did not occur. Confocal settings were held constant across all of the genotypes being tested in an experiment. Maximum z-stack projections were

created by ImageJ (National Institutes of Health). No modifications to any images were made prior to quantification. Larval NMJ4 boutons in segments A2-A4 were manually counted using an Axioplan 2 equipped with a Colibri 2 illumination system using the ×100 1.3 NA oil-immersion objective.

Synapses, defined as the tight apposition between Brp and GluRIII puncta, were manually scored at NMJ4 terminal boutons. Synapse density was calculated by the number of Brp+ synapses per bouton area in $\mu m^2$. The size of DGluRIII clusters was determined by outlining planar DGluRIII clusters in ImageJ and measuring the area. For all Brp+ synapse density and DGluRIII size quantifications, $n$ is the number of NMJ4 boutons scored.

The diameters of isolated planar-oriented Brp rings were measured from confocal images of Brp-stained terminal boutons at NMJ4 deconvolved using Zeiss ZEN software. Deconvolution was performed using the ZEN 2 blue edition default algorithm, the regularized inverse filter, which divides the Fourier transformation of the volume by the Fourier transformation of the point spread function and applies a general cross validation statistical method for the regularization. A line was manually drawn through each Brp ring in ImageJ, and the Brp mean pixel intensity was plotted across the line. The distance from intensity maximum to intensity maximum was determined in the plot window, and this defined as the Brp ring diameter. For all Brp ring analyses, $n$ is the number of NMJ4 boutons scored.

DVGLUT localization was assessed by manually drawing a line through each bouton in ImageJ and DVGLUT mean pixel intensity was plotted across the line. If there were two peaks in the expression plot, and the trough of the plot was less than half of the smallest peak, the bouton was scored as having properly localized DVGLUT. We performed this analysis on boutons larger than 2 μm in diameter, as smaller boutons were often completely filled with synaptic vesicles. The percentage of boutons with properly localized synaptic vesicles relative to the total number of boutons analyzed at single NMJ4 was calculated. For all DVGLUT localization analyses, $n$ is the number of NMJs scored. We measured the integrated pixel intensity over the same drawn line as an estimate of the number of synaptic vesicles. For all integrated DVGLUT analyses, $n$ is the number of NMJ4 boutons scored.

Boutons at the embryonic NMJ were defined as round structures outlined by HRP with a diameter larger than 1 μm. The diameter was measured by drawing two perpendicular lines through each bouton in ImageJ. If both lines were larger than 1 μm, the bouton was counted. For all embryonic analyses, $n$ is the number of NMJ6/7s analyzed.

To determine pMad intensity within motor neuron nuclei, VNCs were imaged at the same confocal settings and equal thickness maximum projections of the dorsal portion of the VNCs were made for each. Regions of interest were determined using Eve immunoreactivity. For each Eve+ motor neuron nucleus within a single VNC, the pMad and Eve pixel intensities were measured. After subtracting background immunofluorescence from each channel, the pMad intensity was divided by the Eve intensity, providing a ratio of pMad to Eve intensity for each Eve+ motor neuron in a single VNC. Ratios were then averaged for each genotype. For the pMad analysis, $n$ is the number of motor neuron cells scored.

To quantify intracellular Gbb-HA, mean pixel intensity was determined inside the inner neuronal membrane for samples incubated in either the 2 mM or 90 mM $K^+$ buffer. Background HA fluorescence was averaged and subtracted from each projection. The neuronal membrane as defined by HRP staining is roughly 200 nm itself. To calculate Gbb-HA proximal to the neuronal membrane, Gbb-HA mean pixel intensity was measured between the intracellular face of the presynaptic membrane as labeled by HRP and a distance of 200 nm away from the extracellular face of the neuronal membrane, a distance of 400 nm in total. Again, background HA fluorescence was averaged and subtracted from each projection. Samples were analyzed again in a similar manner, but the Gbb-HA mean pixel intensity was measured between the intracellular face and extracellular face of the HRP-labeled presynaptic membrane, a total distance of 200 nm. For all Gbb-HA analysis, $n$ is the number of NMJ4s scored.

**Electron microscopy and analysis.** Third-instar larvae were dissected and fixed for 2 h in 2% glutaraldehyde and 2.5% paraformaldehyde. Following post-fixation in 1% osmium tetroxide, the larvae were dehydrated through a graded ethanol series and embedded in Eponate 12. The block was sectioned with a Leica EM UC6 ultramicrotome and stained with uranyl acetate and lead citrate. Images were collected on a FEI Tecnai G2 Spirit BioTWIN transmission electron microscope at the Cleveland Clinic. Electron micrographs of NMJs were taken from NMJ 6/7 and segments A2-A3 in at least two larvae of each genotype. Active zones were identified as linear electron densities found between pre- and postsynaptic membranes. High-magnification images (>125,000) were used to determine the presence of membrane ruffling and/or a T-bar, which was defined as an electron-dense rod surrounded by vesicles and localized to the presynaptic membrane.

**Electrophysiology.** Neuromuscular junction sharp electrode recordings were performed as previously described[74]. Male third-instar larvae were dissected in 0.25 mM $Ca^{2+}$-modified HL3 (70 mM NaCl, 5 mM KCl, 15 mM $MgCl_2$, 10 mM $NaHCO_3$, 115 mM sucrose, 5 mM trehalose, 5 mM HEPES, pH 7.2). Recordings were performed in 0.6 mM $Ca^{2+}$-modified HL3 at muscle 6 of abdominal segments A3 and A4 using sharp borosilicate electrodes filled with 3 M KCl. Recordings were

conducted on a Nikon FN1 microscope using a 40 × 0.80 NA water-dipping objective and acquired using an Axoclamp 900A amplifier, Digidata 1550B acquisition system, and pClamp 11.0.3 software (Molecular Devices). Electrophysiological sweeps were digitized at 10 kHz and filtered at 1 kHz.

For each cell, mean miniature excitatory junctional potential (mEJP) amplitudes were calculated using Mini Analysis (Synaptosoft) from all consecutive events during a one-minute recording in the absence of stimulation. EJPs were stimulated at a frequency of 0.2 Hz using an isolated pulse stimulator 2100 (A-M Systems), with intensity adjusted to consistently elicit compound responses from both type Ib and Is motor neurons. At least 25 consecutive EJPs were recorded for each cell and analyzed in pClamp to obtain mean amplitude. Quantal content was calculated for each recording using the ratio of mean EJP amplitude to mean mEJP amplitude. Recordings were limited to cells with a resting membrane potential between −50 and −85 mV and muscle input resistance greater than 5 MΩ.

**Locomotion assay.** Prior to imaging, 150 mm Petri dishes were prepared, each with a thin layer of 1% agarose and black fountain pen ink (Pelikan) as a dye. L3 larvae were grown at 25 °C and 4–7 larvae were then transferred to the middle of a Petri dish with a few drops of $dH_2O$ to keep the larvae moist. Larvae were imaged for 3 min. Movies were converted to 0.1 Hz (0.1 frames per second) and then larval crawling distance was measured every 10 s using the ImageJ plugin Manual Tracking. $n$ is the total number of larvae scored per genotype in over four experimental trials.

**Statistical analyses.** All statistical analyses were performed using Prism 6 (GraphPad Software). In bar graphs, error bars are mean ± SEM. In the box-and-whisker plots, the center line indicates the median, the boxes cover the interquartile range, and the whiskers show the full range of values. In the scatter plots, the center line represents the mean, and the error bars are mean ± SEM. The $n$ value for each genotype is located at the bottom of each plot. All pairwise comparisons were made using a two-tailed Mann–Whitney test. For determining significance between groups of three or more genotypes, a one-way ANOVA followed by Dunnett's post hoc test was applied on normally distributed data. A nonparametric Kruskal–Wallis one-way ANOVA on ranks followed by a Dunn's multiple comparison test was performed on datasets that did not have a normal distribution. Analyses were conducted with the relevant Gal4 lines or heterozygotes as controls. All measurements were performed on distinct samples without analyzing repeatedly, unless otherwise noted, such as in the Brp ring, DGluRIII cluster size, and integrated DVGLUT analysis. In all figures, $p$ values are as follows: $*p < 0.05$, $**p < 0.01$, and $***p < 0.001$. No significant difference is denoted as n.s. For all figures, source data are provided as a Source Data file.

**Reporting Summary.** Further information on research design is available in the Nature Research Reporting Summary linked to this article.

## Data availability
The data that support the findings of this study are available from the corresponding author upon reasonable request. The source data underlying all plots in Figs. 1I, 2K–L, U–V, 3K–L, 4E–G, 5D, E, I–J, 6F–G, D, I–J, N–S, 8D, H–I, M, Q, 9C, and 10G–H, as well as Supplementary Figs. 1H–I, 3D, H–I, M, 4E, 5I–K, 6D, H–I, M, and 8A, are provided as a Source Data file.

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

## Acknowledgements
We are indebted to Midori Hitomi at the Cleveland Clinic Imaging Core for her electron microscopy expertise and technical assistance. We thank Swati Banerjee, Hugo Bellen, Aaron DiAntonio, Troy Littleton, Thomas Schwarz, Jim Skeath, and Chunlai Wu for fly strains and/or reagents. We also thank the Developmental Studies Hybridoma Bank for antibodies and the Bloomington Drosophila Stock Center for fly stocks. We thank Colleen McLaughlin for thoughtful discussions and feedback. We are grateful for the helpful comments from Pola Philippidou and Dan Jindal on the manuscript. This work was supported by National Institutes of Health (NIH) grants F31NS101763 to K.M.H., R01NS078179 to K.M.O.C.-G., and R01NS095895 to H.T.B. K.M.H. also received support from NIH grant T32GM008056 awarded to the Cell and Molecular Biology Training Program at Case Western Reserve University.

## Author contributions
K.M.H., S.J.G., K.M.O.C.-G., and H.T.B. designed the study, K.M.H., S.J.G., N.Q., K.A.H., Y.L., J.J.P.-R., and P.J.V. conducted experiments, K.M.H., S.J.G., N.Q., K.A.H., P.J.V., K.M.O.C.-G., and H.T.B analyzed data, and K.M.H. and H.T.B. wrote the manuscript with input from K.M.O.C.-G.

## Competing interests
The authors declare no competing interests.
