## [Peer Review File · Nature Communications]

Reviewers' comments:

Reviewer #1 (Remarks to the Author):

The manuscript by Hoover and colleagues describes the role of Gbb mediated signalling, in determining synapse density, structure, and to some degree function. In addition, the authors find that $\alpha 2\delta$ -3 mutants display similar synaptic defects to those observed in *gbb* mutants and that $\alpha 2\delta$ -3 mutant phenotype could be partially suppressed by over-activating the autocrine BMP signalling. The authors propose that a specific loss of the extracellular $\alpha 2$ domain of $\alpha 2\delta$ -3 generates a Gbb diffusion barrier in the extrasynaptic space, which is involved in promoting BMP signalling and thus link $\alpha 2\delta$ -3 into the structure-function regulating BMP signalling pathway.

Per se, I think this paper presents interesting material (particularly the phenotypic characterization of the *gbb* phenotypes) which after substantial revision might result in a publication in Nat Comm. However, I do have a major concern regarding the specificity of the role which $\alpha 2\delta$ -3 might play for Gbb signalling and protein dynamics. The phenotype which they assign to the lack of presynaptic Gbb (large, overgrown but fewer BRP scaffolds) in at least what I can see from their pictures is equally present in mutants of *syd-1*, *Nlg-1* and *Nrx-1*. At the same time, the relation between $\alpha 2\delta$ -3 and *gbb* is just based on phenotypic description, and a rescue experiment (which leaves open whether they are just observing a "race" between two opposing signaling events). What is left is to document that we are speaking a truly specific interaction between $\alpha 2\delta$ -3 and Gbb to explain the changed Gbb dynamics post KCl treatment. Here I am not sure to what degree the differences they observe justifies to assume a really close mechanistic interaction between the two, Gbb and the $\alpha 2\delta$ -3 protein. Might it not be the changed "assembly history or state" of the NMJ terminals in these mutants being responsible here? They at the least would have to investigate how e.g. *syd-1* mutants (ideally also *nrx-1* and *nlg-1*) would behave in direct comparison to their $\alpha 2\delta$ -3 mutants. In the moment, their title I am sorry to say in my eyes is an overstatement.

Additional suggestions and comments:

1. In order to better understand the mechanistic origin of the the $\alpha 2\delta$ -3 and different *gbb* mutants, I suggest to also quantify i) total amounts of BRP (to understand whether the net transport from the cell body is impaired here), ii) the size of the postsynaptic receptor fields which also would give a proxy for PSD size (here it seems to me from the pictures that receptor fields might be increased?). It would be interesting to see how GluRIIA/IIB levels are changed here, again also when comparing $\alpha 2\delta$ -3 and *gbb* mutants.

2. Concerning Fig.3: I am not really sure what to make from the "%boutons with properly localized SVs". To arrive at a more objectified perspective they might like to investigate the cytoskeleton here (e.g. via Futsch stainings), maybe the first candidate to explain these SV distribution changes. dVGLUT images of *Mad12/1* mutant appear to have boutons fully filled with signal and suggesting a complete loss of dVGLUT proper localization and appearing as the strongest phenotype although the data suggests only an intermediate 48% normal SSV distribution in this mutant. The authors might like to look at these data again.

3. Concerning Fig.4: I do not think that the synapse organizational phenotypes they report are able to explain the severe release phenotype of *gbb1/2* animals. What do they think is the ultimate reason for this severe release phenotype (which somewhat surprisingly is not mirrored in the compartment specific KDs)? How about *Cac* mislocalization in *gbb* mutants? Such a defect could establish more common ground with the $\alpha 2\delta$ -3 phenotype. How about $\alpha 2\delta$ -3 physiology?

4. Concerning Fig. 6: The reviewer knows that embryonic NMJ preps are difficult and does think that they did a good job here. However, counting boutons is in my eyes quite subjective, and

counting them by myself in their pictures would not arrive at such a strong reduction. Would there be a phenotype if the counted BRP+ active zones as well? In my eyes, given that they address assembly mechanisms for individual "bouton numbers"

5. For the automated mathematical deconvolution analysis (Figure 2 and 7): there are details missing explaining how many planar AZs per condition were analysed, what averaging and mathematical deconvolution parameters were employed, what the variability in diameter size is in this population. Such data should be presented as non-deconvolved images along-side the deconvolved data with a deconvolution control (eg: for an AZ protein that does not localize as a ring within the AZ such as Cac).

6. concerning Fig. 2: please provide full images of Figure 2O and S to demonstrate the localization of the floating T-bars in *Gbb*^{-/-} mutants with respect to the AZ membrane. Additionally, the dense structure in Figure 2O appears to be at least double the size of wild type T-bar and it would benefit the reader to understand the size and frequency of such floating T-bars occurring/bouton. Linked to this observations, the authors also observe but do not address the unopposed Brp +puncta in Figure 1 and Figure 5 in Neuronal *Gbb*^{-/-}, *Mad*^{-/-} and *wit*^{-/-} mutants? What is the frequency of this occurrence and are they the floating T-bars observed in EM and where are they localized in the bouton? Were similar unopposed AZs observed in $\alpha 2\delta$ -3 mutants?

7. The authors have shown in Figure S4 an impressive and almost entire loss of Cac channels in the $\alpha 2\delta$ -3DD106/K mutants. In addition, Figure 9 shows the specific inability of $\alpha 2\delta$ -3DD106/K mutants to corral *Gbb* in the synaptic cleft to mediate extracellular adhesion or BMP signalling via its lost extracellular $\alpha 2$ domain. How can they tell that the severe effect of $\alpha 2\delta$ -3DD106/K on Cac delocalization is independent from the *Gbb* sequestering effect observed in the same $\alpha 2\delta$ -3DD106/K mutant?

Reviewer #2 (Remarks to the Author):

The manuscript by Hoover et al. reports evidence for how autocrine BMP signaling could impinge upon synaptic organization and function at the *Drosophila* NMJ. The authors also demonstrate that neuronal BMP signaling components appear to work in a shared process or pathway with the $\alpha 2\delta$ (herein, $\alpha 2d$) subunit of presynaptic voltage-gated calcium channels.

Drosophila neuroscientists have long known that BMP signaling plays critical roles in various aspects of NMJ function and development, including synapse growth, neurotransmission, and homeostatic plasticity. It is known that the BMP ligand Glass Bottom Boat (*Gbb*) plays various roles in these processes, either as a retrograde signaling molecule or as a signal originating from the neuron. The current study does a great deal to elucidate autocrine vs. paracrine signaling roles of BMP at the synapse, and the data are very helpful to clarify some puzzles that have hung over the field for some time.

The quality of the experimental data is high. The main assays are synapse imaging (immunofluorescence and EM), but authors have added other approaches to great effect (*Drosophila* genetics, physiology, behavior). The authors make conclusions by extensive genetic analyses, including single mutants, double heterozygotes, presynaptic transgenes, postsynaptic transgenes, etc. The authors also posit a plausible link between BMP signaling and $\alpha 2d$ function, and their data are helpful to the field.

By this reviewer's opinion, there some matters concerning genetic data interpretation that need to be addressed.

MAIN POINTS

1. Pathway Interpretations. This is the main sticking point for this reviewer. There are occasions where the text and figure legends seem to conclude a linear autocrine signaling pathway even though the accompanying genetic data are consistent with other models. On the one hand, this criticism seems semantic – especially since Gbb, Wit, and Mad are well-established members of BMP signaling. However, it is a significant consideration because a running theme of the paper (and the title) is that autocrine BMP signaling is driving synapse organization and function.

- Currently, no double mutant or dual gene combinations are analyzed for Figures 1-6, so the effects of *gbb*, *wit*, and *mad* manipulations could (in theory) be due to parallel effects. It is not quite enough to have shared phenotypes. Regarding an example of shared phenotypes, the authors correctly point out that that losses of *neurexin* gene function also show the SSV distribution phenotype from Figure 3, but there is no thought that *Neurexin* is participating in BMP signaling (correctly so).

- For Figure 7, double heterozygous experiments are suggestive, but by classical genetic analyses double heterozygous phenotypes can be consistent with either linear or parallel pathways.

One remedy would be to soften conclusions throughout the study (and the title itself) and state that neuronal/presynaptic Gbb and Wit are required for these synaptic functions, consistent with a proposed autocrine pathway.

An experimental solution would be to generate *gbb1/2; witA12/B11* double nulls (other double null combinations have been reported as survivable). With double nulls, the prediction would be that the observed phenotypes that are neuron-specific would not be additive. If by some (difficult) trick the authors were able to generate *gbb1/2; witA12/B11* double nulls with *D42-Gal4>UAS-gbb* expressed in this background, the prediction would be non-rescue.

To be absolutely clear: the double null experimental suggestion is NOT to re-do the entire phenotypic analyses in Figures 1-6, only key chosen assays that could test the autocrine BMP pathway.

2. The role of *a2d* is not certain. The data in Figures 7-9 are suggestive of interplay between BMP signaling components and *a2d*. The “physical barrier” description at the end of the results is possible. There are puzzles:

- Pathway analysis. See above – a double null with *a2d* mutants is not possible. Still, if one generated *a2dDD106/k; witA12/B11* doubles, then the phenotype should be no worse than the *wit* null alone (a double using a *gbb1/2* combo would work, but more difficult to recombine).

- Figure 7: It is good that the authors conducted all double het combinations. It is unclear why “genetic redundancy” appears with the *gbb, +/+*, *a2d* double het, but not for other combinations.

- Figure 8: This experiment does not cleanly differentiate between the two possibilities (that *a2d* is either playing an instructive or a permissive role in autocrine BMP signaling). The reason why is because the *DD106/k* combination is hypomorphic, not null.

- Figure 9: This experiment is a great idea, but as the section is written, the data are difficult to interpret. Mainly, how was the 400-nm margin chosen to be an ideal border?

3. Gbb as an autocrine signal vs. a retrograde signal. It is difficult to understand why *BG57-Gal4>UAS-gbb* never rescued any of the phenotypes. Transgenic Gbb protein from the muscle should be released into the synaptic space. As the authors point out in the Discussion (“Multiple BMP signaling pathways...”), the endogenous timing requirement of Gbb does not fully explain things because *BG57* expression lasts for a long time. Regarding their interpretation, I do not quite

understand why the temporal dynamics ("pulse" of Gbb) make the presynaptic-derived signal more robust. In the case of D42-Gal4>UAS-gbb, the transgenic Gbb is presumably constitutively expressed, not pulsed, and that transgenic combination rescues well.

Maybe an alternative way to consider this issue is through the experiment in Figure 9? Could it be the case that muscle-expressed Gbb (if BG57>Gbb-HA were to be tried, for example) is not able to get close enough to where it needs to be (or it is not concentrated enough) to activate the signals governing synaptic organization?

MINOR POINTS

1. Figures 2P-S; Figures 3C,D; Figure S2C. The genotypes of these conditions should be specified in some way (either on the figure or in the legend).
2. Figures 3J and 8H. Is the answer for each DVGLUT analysis a yes/no answer? It seems like there might be graded ways to do this analysis – i.e., if there are two peaks and one of them is 2.1 times the minimum peak and the other is only 1.9 times the minimum peak, does that get coded as "no"?
3. Figure 6. This is a somewhat confusing link to make between the autocrine BMP pathway that is being studied and a2d. We learned in Figure 5 that the synaptic bouton growth phenotype (presumably 3rd instar, muscle 4, type 1b boutons) goes hand in hand with postsynaptic BMP signaling loss. But now a link is made in Figure 6 to synaptic bouton growth defects in embryos, due to presynaptic a2d loss.
4. Figure 8. Given that physiology was done in Figure 4, it is unclear why the locomotion assay is preferred. There is nothing wrong with including the locomotion assay, but there are many neuronal components to locomotion.

Reviewer #3 (Remarks to the Author):

This is a very well written and rigorous manuscript that examines the molecular determinants of synapse development at the *Drosophila* NMJ. The authors find that then BMP signaling molecule Gbb is important for synapse development, and they present evidence that the calcium channel alpha 2 delta 3 subunit is a key modulator of these effects. I have no major technical concerns, with this work, but have a few comments/questions:

Major:

The authors need to demonstrate a biochemical interaction (direct or indirect) between Gbb and alpha2delta3 - if no interaction can be demonstrated, then the model may have to be revised.

Does Gbb overexpression normalize EJP defects in NMJs with mutant alpha2 delta?

Does Gbb acting on alpha2delta3 affect calcium currents?

Minor:

There are two studies that may be relevant for the discussion - Geisler et al J Neuroscience 2019 with regard to the role of alpha2 delta in synaptic architecture, and Gjelsvik et al Molecular Brain 2018 who implicated Gbb signalling in allodynia in *Drosophila*.

BMP is not defined in the text (this could be done in the abstract) or by spelling it out in the title

Response to the reviewers (Nature Communications-19-14451)

We would like to thank the reviewers for their thoughtful and thorough consideration of this work. All three raised important concerns and suggested excellent experiments. In response to their comments, we undertook many additional analyses that have significantly strengthened the manuscript. We highlight a few particularly important revisions here, with detailed point-by-point responses following below.

In response to concerns about the specificity of the $\alpha 2\delta$ -3-Gbb interaction raised by Reviewer 1, we evaluated synaptic phenotypes of *Syd-1* and *Nrx-1* mutants. While both genes are required for synapse organization, their LOF phenotypes are different than those of $\alpha 2\delta$ -3 and Gbb. For example, we now show that Gbb, Wit, and Mad are required for localizing Cac to active zones, while *Nrx-1* and *Syd-1* are not. We went on to test whether either *Nrx-1* or *Syd-1* display dominant genetic interactions with $\alpha 2\delta$ -3 or *gbb*, and find no synapse phenotypes in any of the four different double heterozygote combinations. As described below, this genetic evidence is supported by a strong positive signal in a Proximity Ligation Assay which provides evidence that $\alpha 2\delta$ -3 and motor neuron-derived Gbb are located within a few tens of nanometers of each other. Together, these new analyses indicate that there is an intimate relationship between $\alpha 2\delta$ -3 and Gbb.

In response to concerns about pathway analysis raised by Reviewer 2, we now analyze synapse organization (Brp density/size/multimerization and SSV localization) in animals with presynaptic *gbb* knockdown in a *wit* LOF background (*gbb*^{1/+}; *D42>gbb RNAi*, *wit*^{A12/B11}). We find that *gbb* and *wit* phenotypes are not additive, in line with a linear pathway. We also generated $\alpha 2\delta$ -3^{DD106/k}; *wit*^{A12/B11} double mutants, and find that the double mutant phenotype is no more severe than either single mutant. Together, these analyses support the conclusion that $\alpha 2\delta$ -3 facilitates autocrine BMP signaling.

In response to a question regarding a possible physical interaction between $\alpha 2\delta$ -3 and Gbb raised by Reviewer 3, we have added an entirely new figure (Figure 9). Hugo Bellen generously provided $\alpha 2\delta$ -3 antibody generated in his lab. We validated the reagent in our hands, and then used it to demonstrate that $\alpha 2\delta$ -3 and motor neuron-derived Gbb extensively colocalize in the synaptic cleft. We then employed a Proximity Ligation Assay to test whether the proteins are situated closely enough to generate a positive signal in this assay (<40 nm) and detect a robust PLA signal. Together, these analyses are consistent with a physical interaction.

We have highlighted significantly changed text in the revised manuscript, and reference the relevant figures in our response below.

Reviewer #1

Major Point:

I do have a major concern regarding the specificity of the role which $\alpha 2\delta$ -3 might play for Gbb signaling and protein dynamics. The phenotype which they assign to

the lack of presynaptic Gbb (large, overgrown but fewer BRP scaffolds) in at least what I can see from their pictures is equally present in mutants of *syd-1*, *Nlg-1* and *Nrx-1*. At the same time, the relation between $\alpha 2\delta$ -3 and *gbb* is just based on phenotypic description, and a rescue experiment (which leaves open whether they are just observing a “race” between two opposing signaling events). What is left is to document that we are speaking a truly specific interaction between $\alpha 2\delta$ -3 and Gbb to explain the changed Gbb dynamics post KCl treatment. Here I am not sure to what degree the differences they observe justifies to assume a really close mechanistic interaction between the two, Gbb and the $\alpha 2\delta$ -3 protein. Might it not be the changed “assembly history or state” of the NMJ terminals in these mutants being responsible here? They at the least would have to investigate how e.g. *syd-1* mutants (ideally also *nrx-1* and *nlg-1*) would behave in direct comparison to their $\alpha 2\delta$ -3 mutants. In the moment, their title I am sorry to say in my eyes is an overstatement.

This key comment articulates the need to demonstrate specificity of the presynaptic Gbb- $\alpha 2\delta$ -3 interaction, and we undertook many experiments to address it:

1. As suggested, we repeated the phenotypic analysis of Brp density/size/multimerization and SSV localization in *Syd-1* and *Nrx-1* mutants. As the reviewer correctly points out, these phenotypes are similar to those described here for Gbb and $\alpha 2\delta$ -3; however, we also find significant differences (**Figure S3**). Moreover, we now also include an analysis of Cac localization not only in *gbb*, *wit*, and *Mad* mutants (as suggested by this reviewer), but also in *Nrx-1* and *Syd-1* mutants. We find that while Gbb signaling is required for Cac localization—*Nrx-1* and *Syd-1* are not (**Figure S5 A-I**). Together, these findings reinforce the phenotypic similarities between Gbb and $\alpha 2\delta$ -3.

2. To evaluate the specificity of the genetic interaction between $\alpha 2\delta$ -3 and Gbb, we tested whether either gene interacts with *Syd-1* or *Nrx-1*. We find that both Brp density and SSV localization are normal in these four different double heterozygotes (**Figure S5 J-K**). These results provide strong evidence of a selective genetic interaction between $\alpha 2\delta$ -3 and Gbb.

3. We generated *a2 δ -3*; *wit* double homozygotes, and find that the phenotype of this double mutant is no more severe than that of either single mutant (**Figure 7 R-S**), providing key genetic evidence that $\alpha 2\delta$ -3 facilitates BMP signaling. (We used *wit* here since *a2 δ -3* and *gbb* are on the same chromosome.)

4. We now show that overexpression of Gbb from the post-synaptic side is not sufficient to rescue $\alpha 2\delta$ -3 mutant phenotypes (**Figure S6 A-M**). Thus, the ability to rescue $\alpha 2\delta$ -3 mutant phenotypes is specific to pre- (and not post-) synaptic Gbb.

5. Using an $\alpha 2\delta$ -3 antibody kindly provided by Hugo Bellen, we now demonstrate that $\alpha 2\delta$ -3 and presynaptic BMP co-localize extensively (**Figure 9 D**). We employ a PLA (Proximity Ligation Assay) to demonstrate that the two proteins are in close (<40 nm) proximity in the extracellular space (**Figure 9 E**).

Together, these experiments provide strong support for a specific interaction between presynaptic Gbb and $\alpha 2\delta$ -3.

Additional suggestions and comments:

1. In order to better understand the mechanistic origin of the the $\alpha 2\delta$ -3 and different *gbb* mutants, I suggest to also quantify i) total amounts of BRP (to understand whether the net transport from the cell body is impaired here), ii) the size of the postsynaptic receptor fields which also would give a proxy for PSD size (here it seems to me from the pictures that receptor fields might be increased?). It would be interesting to see how GluRIIA/IIB levels are changed here, again also when comparing $\alpha 2\delta$ -3 and *gbb* mutants.

i) As suggested, we performed Brp immunoblots from *gbb* pathway mutant VNCs. But the experiment did not give a clear result given variability among genotypes. Across multiple experiments, we see unexpected variability (compare wild-type extracts in lanes 1, 2, and 6). While we did not find convincing evidence that Gbb

regulates overall Brp levels, it is still possible that BMP signaling regulates Brp processing/stability/localization. We would like to carefully pursue this question in comprehensive studies in the future.

ii) As suggested, we quantified glutamate receptor field size in BMP pathway mutants. And as this reviewer predicted, postsynaptic receptor field size increases specifically in genotypes lacking presynaptic Gbb signaling (**Figure S1 I**). Thus, this pathway regulates both pre- and postsynaptic differentiation. Moreover, we agree that the question of the GluRIIA/IIB ratio is interesting; however, we feel it is outside the scope of the current work.

2. Concerning Fig.3: I am not really sure what to make from the "%boutons with properly localized SVs". To arrive at a more objectified perspective they might like to investigate the cytoskeleton here (e.g. via Futsch stainings), maybe the first candidate to explain these SV distribution changes. dVGLUT images of *Mad12/1* mutant appear to have boutons fully filled with signal and suggesting a complete loss of dVGLUT proper localization and appearing as the strongest phenotype although the data suggests only an intermediate 48% normal SSV distribution in this mutant. The authors might like to look at these data again.

Regarding the underlying cause of the SSV phenotype, we also expect that cytoskeletal defect(s) may be the culprit. And we are interested in identifying the downstream transcriptional effectors. But we think this is outside the scope of this manuscript.

We thank the reviewer for pointing out that we did not choose a representative example for the *Mad*^{12/1} image. We replaced it with an image that better captures the slightly more intermediate phenotype of this genotype.

3. Concerning Fig.4: I do not think that the synapse organizational phenotypes they report are able to explain the severe release phenotype of *gbb*^{1/2} animals. What do they think is the ultimate reason for this severe release phenotype (which somewhat surprisingly is not mirrored in the compartment specific KDs)? How about Cac mislocalization in *gbb* mutants? Such a defect could establish more common ground with the $\alpha 2\delta$ -3 phenotype. How about $\alpha 2\delta$ -3 physiology?

We think that the severe release phenotype of *gbb*^{1/2} mutants is an additive effect of the drastic reduction in bouton number and synapse organization defects. Our point is that selective loss of presynaptic Gbb does not alter bouton number, but *does* lead to a significant reduction in quantal content.

We thank the reviewer for this idea. We thought the suggestion to look at Cac localization in *gbb* mutants was a great one! We analyzed Cac localization in *gbb*, *wit*, and *Mad* mutants and find that roughly 20% of Brp-positive active zones lack Cac (**Figure S5 D-E, I**). We could not check the compartment-specific *gbb* knockouts because it required generating X; II; III balanced stocks, which we could not keep alive (though we tried). We also evaluated Cac localization in *Nrx-1* and *Syd-1* mutants and do not find a requirement for either protein in localizing Cac to active zones (**Figure S5 G-H, I**). This experiment extends the phenotypic similarity between $\alpha 2\delta$ -3 and presynaptic Gbb signaling to another area of presynaptic assembly.

We did not repeat the electrophysiology that was previously performed in *$\alpha 2\delta$ -3^{DD106/k}* mutants (Dickman et al., 2008).

4. Concerning Fig. 6: The reviewer knows that embryonic NMJ preps are difficult and does think that they did a good job here. However, counting boutons is in my eyes quite subjective, and counting them by myself in their pictures would not arrive at such a strong reduction. Would there be a phenotype if the counted BRP+ active zones as well? In my eyes, given that they address assembly mechanisms for individual “bouton numbers”

We agree that there is a degree of subjectivity to counting boutons, especially in embryos, where they can be quite small. To reduce subjectivity, we repeated the analysis counting only boutons with a diameter > 1 μ M (**Figure 6 G**). This approach yielded the same result—a significant decrease in bouton number in *$\alpha 2\delta$ -3* and *gbb* pathway mutants.

5. For the automated mathematical deconvolution analysis (Figure 2 and 7): there are details missing explaining how many planar AZs per condition were analysed, what averaging and mathematical deconvolution parameters were employed, what the variability in diameter size is in this population. Such data should be presented as non-deconvolved images along-side the deconvolved data with a

deconvolution control (eg: for an AZ protein that does not localize as a ring within the AZ such as Cac).

We added more detail about this analysis in the Methods, as well as information about how many Brp rings were analyzed per genotype. This information is also available in the Source File submitted with this manuscript, which contains all spread sheets for all experiments. There are error bars in all of these analyses to convey experimental variability. And as suggested, we now provide deconvolved images side-by-side original images (**Figure S2 A-B'**). We also performed the requested control and show that Cac does not appear as a ring following deconvolution (**Figure S2 C-C'**).

6. Concerning Fig. 2: please provide full images of Figure 2O and S to demonstrate the localization of the floating T-bars in *Gbb*^{-/-} mutants with respect to the AZ membrane. Additionally, the dense structure in Figure 2O appears to be at least double the size of wild type T-bar and it would benefit the reader to understand the size and frequency of such floating T-bars occurring/bouton. Linked to this observations, the authors also observe but do not address the unopposed Brp⁺ puncta in Figure 1 and Figure 5 in Neuronal *Gbb*^{-/-}, *Mad*^{-/-} and *wit*^{-/-} mutants? What is the frequency of this occurrence and are they the floating T-bars observed in EM and where are they localized in the bouton? Were similar unopposed AZs observed in $\alpha 2\delta$ -3 mutants?

As requested, we replaced the original images in Figure 2 with new ones representing the entire bouton (**Figure 2 P, T**). As the reviewer notes, some of these aggregates are quite large, which is why they are so obvious in the EM. As requested, we quantified the frequency of the floating T-bar material per bouton, and find that it is between 10-30%. We now speculate that this phenotype may partially explain the unopposed Brp puncta observed at the light level. While we occasionally observe unopposed Brp puncta in $\alpha 2\delta$ -3 mutants, this phenotype is not as obvious as in *gbb* mutants.

7. The authors have shown in Figure S4 an impressive and almost entire loss of Cac channels in the $\alpha 2\delta$ -3DD106/K mutants. In addition, Figure 9 shows the specific inability of $\alpha 2\delta$ -3DD106/K mutants to corral Gbb in the synaptic cleft to mediate extracellular adhesion or BMP signaling via its lost extracellular $\alpha 2$ domain. How can they tell that the severe effect of $\alpha 2\delta$ -3DD106/K on Cac delocalization is independent from the Gbb sequestering effect observed in the same $\alpha 2\delta$ -3DD106/K mutant?

The short answer is that we can't entirely rule it out, at least at this point. But we think that the function of $\alpha 2\delta$ -3 in localizing Cac channels is largely independent of Gbb. This is supported by our comparative analysis of $\alpha 2\delta$ -3^{106/k} vs. $\alpha 2\delta$ -3^{196/k} mutants. Both alleles severely disrupt Cac localization, but only 106/k disrupts Gbb signaling and synapse organization. 196/k, which has an intact $\alpha 2$ domain, is not required for synapse assembly, but does regulate Cac localization. This argues that $\alpha 2\delta$ -3 serves a function

in Cac localization independent of its function in BMP signaling. In the future, we are interested in carefully parsing apart the relative roles of $\alpha 2\delta$ -3 in Ca channel localization vs. BMP signaling, but we think that is beyond the current scope.

Reviewer #2:

MAIN POINTS

1. Pathway Interpretations. This is the main sticking point for this reviewer. There are occasions where the text and figure legends seem to conclude a linear autocrine signaling pathway even though the accompanying genetic data are consistent with other models. On the one hand, this criticism seems semantic – especially since Gbb, Wit, and Mad are well-established members of BMP signaling. However, it is a significant consideration because a running theme of the paper (and the title) is that autocrine BMP signaling is driving synapse organization and function.

• **Currently, no double mutant or dual gene combinations are analyzed for Figures 1-6, so the effects of gbb, wit, and mad manipulations could (in theory) be due to parallel effects. It is not quite enough to have shared phenotypes. Regarding an example of shared phenotypes, the authors correctly point out that that losses of neurexin gene function also show the SSV distribution phenotype from Figure 3, but there is no thought that Neurexin is participating in BMP signaling (correctly so).**

• **For Figure 7, double heterozygous experiments are suggestive, but by classical genetic analyses double heterozygous phenotypes can be consistent with either linear or parallel pathways.**

One remedy would be to soften conclusions throughout the study (and the title itself) and state that neuronal/presynaptic Gbb and Wit are required for these synaptic functions, consistent with a proposed autocrine pathway.

An experimental solution would be to generate gbb1/2; wit^{A12/B11} double nulls (other double null combinations have been reported as survivable). With double nulls, the prediction would be that the observed phenotypes that are neuron-specific would not be additive. If by some (difficult) trick the authors were able to generate gbb1/2; wit^{A12/B11} double nulls with D42-Gal4>UAS-gbb expressed in this background, the prediction would be non-rescue.

To be absolutely clear: the double null experimental suggestion is NOT to re-do the entire phenotypic analyses in Figures 1-6, only key chosen assays that could test the autocrine BMP pathway.

We appreciate this concern, and agree that further experimentation was warranted given our emphasis on the autocrine pathway. We generated a *gbb*^{1/2}; *wit*^{A12/B11} stock,

but despite our best efforts, the double homozygotes died as first instar larvae. As an alternative, we generated animals with presynaptic *gbb* knockdown in a *wit* background (*gbb*^{1/+}; *D42>gbb RNAi*, *wit*^{A12/B11}), and found that these animals survive to thirds. We analyzed Brp density/size/multimerization and SSV distribution in this genotype, and find that the *gbb* and *wit* phenotypes are not additive (**Figure 1 G, I; Figure 2 G, K, L; Figure 3 G, K**). These data support the proposed autocrine pathway.

2. The role of a2d is not certain. The data in Figures 7-9 are suggestive of interplay between BMP signaling components and a2d. The “physical barrier” description at the end of the results is possible. There are puzzles:

• **Pathway analysis. See above – a double null with a2d mutants is not possible. Still, if one generated a2dDD106/k; witA12/B11 doubles, then the phenotype should be no worse than the wit null alone (a double using a gbb1/2 combo would work, but more difficult to recombine).**

This was a great suggestion. We generated the *a2δ-3*^{106/k}; *wit*^{A12/B11} double homozygotes and find that the synapse organization phenotypes are no more severe than in either single mutant (**Figure 7 R, S**).

• **Figure 7: It is good that the authors conducted all double het combinations. It is unclear why “genetic redundancy” appears with the *gbb*, +/+, a2d double het, but not for other combinations.**

We didn't understand why we initially observed redundancy in *gbb*¹, *a2δ-3*¹⁰⁶ hets for the SSV phenotype either, but wondered if it was attributable to the relatively small sample size. We repeated the experiment for a larger sample size (n=24, similar to the number we scored for other genotypes), and now see a significant interaction (**Figure 7 Q**), though still not as striking as the *a2δ-3*; *wit* interaction. We always saw interactions between *a2δ-3* and *gbb* in the synapse density assay (**Figure 7 P**), and taken with the strong dominant genetic interactions between *a2δ-3* and *wit*, as well as the newly generated *a2δ-3* *wit* double homozygote—we think these data support our finding that *a2δ-3* contributes to the autocrine BMP pathway.

• **Figure 8: This experiment does not cleanly differentiate between the two possibilities (that a2d is either playing an instructive or a permissive role in autocrine BMP signaling). The reason why is because the DD106/k combination is hypomorphic, not null.**

Agreed. We did not mean to imply that presynaptic Gbb's ability to suppress synapse organization defects in *a2δ-3* mutants definitively rules out an instructive role for *a2δ-3* in the Gbb pathway, and we have softened the language throughout this section. It is possible that residual *a2δ-3* function in the DD106/k background is required for Gbb's activity in this assay. To pursue this question, we could test whether *a2δ-3* modulates

BMP receptor activity/localization or perhaps Gbb processing/stability in addition to its role in regulating Gbb localization in the synaptic cleft. But we think these experiments are beyond the scope of the current manuscript.

• **Figure 9: This experiment is a great idea, but as the section is written, the data are difficult to interpret. Mainly, how was the 400-nm margin chosen to be an ideal border?**

We chose 400 nm because (1) it captured roughly equal parts presynaptic membrane and extracellular space and (2) and we were confident we could consistently define a region of this width in ImageJ. In response to this concern, we repeated the analysis tightening the margin to 200 nm, which is roughly the resolution limit of the confocal, and observed exactly the same result (**Figure S8 A**).

3. Gbb as an autocrine signal vs. a retrograde signal. It is difficult to understand why BG57-Gal4>UAS-gbb never rescued any of the phenotypes. Transgenic Gbb protein from the muscle should be released into the synaptic space. As the authors point out in the Discussion (“Multiple BMP signaling pathways...”), the endogenous timing requirement of Gbb does not fully explain things because BG57 expression lasts for a long time. Regarding their interpretation, I do not quite understand why the temporal dynamics (“pulse” of Gbb) make the presynaptic-derived signal more robust. In the case of D42-Gal4>UAS-gbb, the transgenic Gbb is presumably constitutively expressed, not pulsed, and that transgenic combination rescues well.

Maybe an alternative way to consider this issue is through the experiment in Figure 9? Could it be the case that muscle-expressed Gbb (if BG57>Gbb-HA were to be tried, for example) is not able to get close enough to where it needs to be (or it is not concentrated enough) to activate the signals governing synaptic organization?

We agree that this is a key underlying mystery, and one that we continue to explore. Even when we drive Gbb on the post-synaptic side through larval development, it does not rescue synapse organization, consistent with the finding that sustained muscle expression of Gbb does not rescue baseline release in *gbb* nulls (Goold and Davis, 2007; McCabe et al., 2003). Given other studies of BMP receptor activation (Sorre et al., 2014; Warmflash et al., 2012), we suspect that the relative timing of Gbb release from MN vs muscle is important. Neuronal Gbb is not in the constitutive secretory pathway, but rather is in DCVs and released following neuronal activity (James et al., 2014). In the absence of activity, extracellular levels of neuronal Gbb are low (**Figure 10 A”-B”**). It may be the timing of Gbb release (or its coupling to activity) that is required for efficient receptor activation. In contrast, we have no evidence that muscle-derived Gbb release is regulated by activity. We did not make this idea clear in the original submission but have tried to do a better job articulating it in the revised Discussion.

MINOR POINTS

1. Figures 2P-S; Figures 3C,D; Figure S2C. The genotypes of these conditions should be specified in some way (either on the figure or in the legend).

We apologize we did not originally spell these out. They are now defined in all the relevant figure legends.

2. Figures 3J and 8H. Is the answer for each DVGLUT analysis a yes/no answer? It seems like there might be graded ways to do this analysis – i.e., if there are two peaks and one of them is 2.1 times the minimum peak and the other is only 1.9 times the minimum peak, does that get coded as “no”?

Yes, we used the lower of the two peaks for the SSV analysis. We scored as “properly localized” those boutons where the lower peak was $>2.0X$ the minimum. (We now spell this out in the Materials & Methods). So in this reviewer’s hypothetical, the bouton is scored as a “no”. Certainly, there are other ways to do this analysis. We would point out that the group that initially characterized this SSV phenotype in *Nrx-1* mutants visually sorted boutons into “localized” and “diffuse” bins (Rui et al., 2017). We wanted to be able to compare our phenotypes to those described in this paper, which is why we set up the assay as we did. And to make it quantitative, we set the $2.0X$ threshold. This mutant phenotype is obvious and robust. For example, we repeated it on a relatively small number of boutons from a few control and *gbb* mutant backgrounds employing a $1.9X$ threshold and observe the same result (see attached figure).

3. Figure 6. This is a somewhat confusing link to make between the autocrine BMP pathway that is being studied and a2d. We learned in Figure 5 that the synaptic bouton growth phenotype (presumably 3rd instar, muscle 4, type 1b boutons) goes hand in hand with postsynaptic BMP signaling loss. But now a link is made in Figure 6 to synaptic bouton growth defects in embryos, due to presynaptic a2d loss.

We totally agree and debated at great length how best to organize this study. It seemed logical that bouton formation would require postsynaptic BMP, since this ligand pool regulates bouton number. And while we didn’t identify genetic conditions to enable compartment-specific experiments in embryos, we used *crimpy* LOF animals to approach the same question. (We previously demonstrated a specific requirement for Crimpy in regulating trafficking and release of presynaptic Gbb (James et al., 2014). Crimpy is not expressed in muscle and does not regulate the function of the postsynaptic pool.) Surprisingly, in *crimpy* homozygous mutant embryos, we see the same deficit in bouton formation as in *gbb* and *wit* mutants. This is why we think

presynaptic Gbb may regulate bouton morphogenesis. We did not originally include the *crimpy* LOF phenotype as the interpretation of later-stage NMJ phenotypes is complicated by the finding that *crimpy* may regulate the trafficking of other growth factors. But we see that without these data, the embryonic phenotypes are particularly confusing. We have now included this analysis, and have tried to be clearer about our reasoning here (**Figure 6 D, G; lines 369-377**).

Finally, it is the embryonic analysis that first suggested the $\alpha 2\delta$ -3 link, which is why we placed it immediately before the $\alpha 2\delta$ -3 section of the paper.

4. Figure 8. Given that physiology was done in Figure 4, it is unclear why the locomotion assay is preferred. There is nothing wrong with including the locomotion assay, but there are many neuronal components to locomotion.

As we were sorting mutant larvae, we had the impression that $\alpha 2\delta$ -3 mutant larvae with presynaptic expression of Gbb were more vigorous than $\alpha 2\delta$ -3 mutants on their own, which was what motivated the analysis. Indeed, overall motor function is modestly improved, though still reduced relative to controls. We were surprised by this result, given the stringent (and likely Gbb-independent) requirement for $\alpha 2\delta$ -3 in Cac localization.

In response to this comment, we made a first pass at comparing baseline ephys in these genotypes and did not see anything obvious. But there are significant differences in the various background controls that may be masking subtle differences in neurotransmission. In the future, we would like to analyze baseline release across different conditions in these backgrounds as well as function in short-term presynaptic plasticity assays to understand the relationship between $\alpha 2\delta$ -3's function in Cac localization vs. Gbb signaling. As these experiments are beyond the scope of the current work, we tried to maintain a focus specifically on the Gbb-related functions of $\alpha 2\delta$ -3 throughout this section of the revised manuscript.

Reviewer #3:

Major:

The authors need to demonstrate a biochemical interaction (direct or indirect) between Gbb and alpha2delta3 - if no interaction can be demonstrated, then the model may have to be revised.

We propose that the extracellular $\alpha 2$ domain of $\alpha 2\delta$ -3 serves as a physical barrier to Gbb diffusion. While we do not think that our model necessitates a stable interaction between Gbb and $\alpha 2\delta$ -3, it is certainly true that the discovery of a $\alpha 2\delta$ -3-Gbb complex would lend support to the idea. Because the proteins are predicted to interact in the ECM of the synaptic cleft, we didn't think that this interaction was likely to be preserved in extracts from cultured cells. Thus, we undertook co-IPs from third instar body wall extracts, but despite significant effort, were unable to detect the proteins in a stable complex. In large part, our efforts were stymied by Gbb's non-specific interactions with

Sepharose 4B beads in a pre-clearing step.

As an alternative means to test if the two proteins are in close (<40 nm) proximity to each other in the extracellular space, we turned to a Proximity Ligation Assay (PLA). PLA relies on standard antibody detection using secondary antibodies coupled to oligonucleotide probes that can ligated together only if the primary antibodies are in close proximity. This ligated product is then detected using a complementary fluorescent probe. This assay is increasingly used to detect protein-protein interactions *in situ* (Lepicard et al., 2014; Mosca et al., 2017; Wang et al., 2015). It is reported to give similar results as co-immunoprecipitation, with the added benefit of relaying spatial information of interacting proteins (Graf et al., 2011; Vizlin-Hodzic et al., 2011). Using an $\alpha 2\delta$ -3 antibody that we obtained from Hugo Bellen, we find a robust PLA signal between $\alpha 2\delta$ -3 and Gbb, consistent with a physical interaction. These data are presented in **Figure 9**, an entirely new figure where we present antibody validation, Gbb/ $\alpha 2\delta$ -3 colocalization, and the PLA experiment. PLA controls are in **Figure S7**. These data argue that $\alpha 2\delta$ -3 is well positioned to promote Gbb retention at the presynaptic membrane.

Does Gbb overexpression normalize EJP defects in NMJs with mutant alpha2 delta?

Please see our response to the 4th Minor Point of Reviewer 2.

Does Gbb acting on alpha2delta3 affect calcium currents?

We are very interested in parsing apart Gbb-dependent and Gbb-independent functions of $\alpha 2\delta$ -3. But we think these studies are beyond the scope of the current manuscript.

Minor:

There are two studies that may be relevant for the discussion - Geisler et al J Neuroscience 2019 with regard to the role of alpha2 delta in synaptic architecture, and Gjelsvik et al Molecular Brain 2018 who implicated Gbb signalling in allodynia in Drosophila.

We now mention Geisler et al. in the introduction when we first set up the widespread synapse organizing function of $\alpha 2\delta$ proteins. We cite Gjelsvik in the Discussion when we summarize the findings that autocrine BMP signaling and $\alpha 2\delta$ -3 have both been linked to pain sensation/sensitivity in flies—raising the possibility that our findings on a functional $\alpha 2\delta$ -3-Gbb interaction may extend to other settings.

BMP is not defined in the text (this could be done in the abstract) or by spelling it out in the title.

We now define BMP in the abstract.

References

- Goold, C.P., and Davis, G.W. (2007). The BMP ligand Gbb gates the expression of synaptic homeostasis independent of synaptic growth control. *Neuron* 56, 109–123.
- Graf, M., Brobeil, A., Sturm, K., Steger, K., and Wimmer, M. (2011). 14-3-3 beta in the healthy and diseased male reproductive system. *Hum. Reprod.* 26, 59–66.
- James, R.E., Hoover, K.M., Bulgari, D., McLaughlin, C.N., Wilson, C.G., Wharton, K.A., Levitan, E.S., and Broihier, H.T. (2014). Crimpy enables discrimination of presynaptic and postsynaptic pools of a BMP at the *Drosophila* neuromuscular junction. *Dev. Cell* 31, 586–598.
- Lepicard, S., Franco, B., de Bock, F., and Parmentier, M.-L. (2014). A presynaptic role of microtubule-associated protein 1/Futsch in *Drosophila*: regulation of active zone number and neurotransmitter release. *J. Neurosci.* 34, 6759–6771.
- McCabe, B.D., Marques, G., Haghighi, A.P., Fetter, R.D., Crotty, M.L., Haerry, T.E., Goodman, C.S., and O'Connor, M.B. (2003). The BMP homolog Gbb provides a retrograde signal that regulates synaptic growth at the *Drosophila* neuromuscular junction. *Neuron* 39, 241–254.
- Mosca, T.J., Luginbuhl, D.J., Wang, I.E., and Luo, L. (2017). Presynaptic LRP4 promotes synapse number and function of excitatory CNS neurons. *Elife* 6, 2185.
- Rui, M., Qian, J., Liu, L., Cai, Y., Lv, H., Han, J., Jia, Z., and Xie, W. (2017). The neuronal protein Neurexin directly interacts with the Scribble-Pix complex to stimulate F-actin assembly for synaptic vesicle clustering. *J. Biol. Chem.* 292, 14334–14348.
- Sorre, B., Warmflash, A., Brivanlou, A.H., and Siggia, E.D. (2014). Encoding of temporal signals by the TGF- β pathway and implications for embryonic patterning. *Dev. Cell* 30, 334–342.
- Vizlin-Hodzic, D., Runnberg, R., Ryme, J., Simonsson, S., and Simonsson, T. (2011). SAF-A forms a complex with BRG1 and both components are required for RNA polymerase II mediated transcription. *PLoS ONE* 6, e28049.
- Wang, S., Yoo, S., Kim, H.-Y., Wang, M., Zheng, C., Parkhouse, W., Krieger, C., and Harden, N. (2015). Detection of in situ protein-protein complexes at the *Drosophila* larval neuromuscular junction using proximity ligation assay. *J Vis Exp* 52139.
- Warmflash, A., Zhang, Q., Sorre, B., Vonica, A., Siggia, E.D., and Brivanlou, A.H. (2012). Dynamics of TGF- β signaling reveal adaptive and pulsatile behaviors reflected in the nuclear localization of transcription factor Smad4. *Proc. Natl. Acad. Sci. U.S.A.* 109, E1947–E1956.

REVIEWERS' COMMENTS:

Reviewer #1 (Remarks to the Author):

I congratulate the authors to a substantial and largely successful revision. Though I still think that ultimate proof of a specific connection between gbb signaling and a2d would need deeper molecular analysis, I consider the revised version an important contribution to the field and now suggest publication as is.

Reviewer #2 (Remarks to the Author):

Hoover et al. is a revision of a prior submission. This manuscript documents the importance of an autocrine, presynaptic BMP signaling pathway and how that signaling is promoted by the $\alpha 2\delta$ -3 calcium channel subunit. To support their conclusions, the authors have combined synapse imaging, electrophysiology, EM, genetics, in vivo biochemistry, and behavior.

From the first round of review, all three reviewers requested difficult and significant experimental revisions to bolster the conclusions of the paper. Reviewer #1 requested more information to support the specificity of the relationship between Gbb and $\alpha 2\delta$ -3. This reviewer (#2) requested several more genetic tests of the linearity of the proposed autocrine pathway. Reviewer #3 requested a biochemical test of a potential Gbb/ $\alpha 2\delta$ -3 interaction.

The authors recognized the validity of each concern and undertook a thorough approach to address all of them. This was obvious simply by an examination of the revised figures. Every major point and most minor points have been addressed in some manner, usually experimentally. For minor points that are not central to the core conclusions, the authors have provided good reasons why and informative discussion. This is a great compendium of work.

Minor Edits/Suggestions for final version

1. Nomenclature: Do not forget to add the crimp genetic information to the methods, and try to make the nrx/Nrx and syd/Syd uses for genotypes consistent with proper capitalization in both the figures and text.
2. For p value instances of n.s. (non-significance), it seems that all values > 0.05 are marked in exactly this way. For full transparency, it might be better to mark instances that are $0.05 < p < 0.1$ with the actual p value (or all instances of n.s. with an actual value or a marker that $p > 0.3$, for example).

Reviewer #3 (Remarks to the Author):

The authors have addressed most of comments - I am fine with this as is

With respect to our final reviewer response, Reviewer 2 had very minor capitalization and/or nomenclature issues, which we fixed.